# Global-scale drought risk assessment for agricultural systems

Isabel Meza[1], Stefan Siebert[2], Petra Döll[3,6], Jürgen Kusche[4], Claudia Herbert[3], Ehsan Eyshi Rezaei[2], Hamideh Nouri[2], Helena Gerdener[4], Eklavyya Popat[3], Janna Frischen[1], Gustavo Naumann[5], Jürgen V. Vogt[5], Yvonne Walz[1], Zita Sebesvari[1], Michael Hagenlocher[1]

[1] United Nations University, Institute for Environment and Human Security (UNU-EHS), UN Campus, Platz der Vereinten Nationen 1, 53113 Bonn, Germany
[2] Department of Crop Sciences, University of Göttingen, Von-Siebold-Strasse 8, 37075 Göttingen, Germany
[3] Institute of Physical Geography, Goethe University Frankfurt, Altenhöferallee 1, 60438 Frankfurt am Main, Germany
[4] Institute of Geodesy and Geoinformation (IGG), University of Bonn, Nussallee 17, 53115 Bonn, Germany
[5] European Commission (EC), Joint Research Centre (JRC), Via Enrico Fermi 2749, 21027 Ispra, VA, Italy
[6] Senckenberg Leibniz Biodiversity and Climate Research Centre Frankfurt (SBiK-F), Senckenberganlage 25, 60325 Frankfurt am Main, Germany

*Correspondence to*: Isabel Meza (meza@ehs.unu.edu)

Authors Orcid iD:
Meza: 0000-0002-8557-9175
Hagenlocher: 000-0002-5254-6713
Siebert: 0000-0002-9998-0672
Sebesvari: 0000-0001-7686-1227
Herbert: 0000-0002-4795-5328
Döll: 0000-0003-2238-4546
Popat: 0000-0002-3064-163X
Vogt: 0000-0003-2955-9484
Gerdener: 0000-0003-1043-1923
Kusche: 0000-0001-7069-021X
Naumann: 0000-0002-8767-5099
Nouri: 0000-0002-7424-5030
Eyshi Rezaei: 0000-0003-2603-8034
Frischen: 0000-0002-6379-7489
Walz: 0000-0003-3781-5038

## Abstract

Droughts continue to affect ecosystems, communities, and entire economies. Agriculture bears much of the impact, and in many countries it is the most heavily affected sector. Over the past decades, efforts have been made to assess drought risk at different spatial scales. Here, we present for the first time an integrated assessment of drought risk for both irrigated and rain-fed agricultural systems at the global scale. Composite hazard indicators were calculated for irrigated and rain-fed systems separately using different drought indices based on historical climate conditions (1980-2016). Exposure was analyzed for

irrigated and non-irrigated crops. Vulnerability was assessed through a social-ecological systems perspective, using social-ecological susceptibility and lack of coping capacity indicators that were weighted by drought experts from around the world. The analysis shows that drought risk of rain-fed and irrigated agricultural systems displays a heterogeneous pattern at the global level with higher risk for southeastern Europe, as well as northern and southern Africa. By providing information on the drivers and spatial patterns of drought risk in all dimensions of hazard, exposure, and vulnerability, the presented analysis can support the identification of tailored measures to reduce drought risk and increase the resilience of agricultural systems.

**Keywords:** Drought, Hazard, Exposure, Vulnerability, Rain-fed agriculture, Irrigated agriculture

## 1 Introduction

Droughts exceed all other natural hazards in terms of the number of people affected, and have contributed to some of the world's most severe famines (FAO, 2018; CRED and UNISDR, 2018). Drought is conceived as an exceptional and sustained lack of water caused by a deviation from normal conditions over a certain region (Tallaksen and Van Lanen, 2004, Van Loon et al., 2016). It can have manifold impacts on social, ecological, and economic systems, for instance agricultural losses, public water shortages, reduced hydropower supply, and reduced labor or productivity. While many sectors are affected by drought, agriculture's high dependency on water means it is often the first one of the most heavily affected sectors (Dilley et al., 2005; UNDRR, 2019). With nearly 1.4 billion people (18% of the global population) employed in agriculture, droughts threaten the livelihoods of many, and are hampering the achievement of the Sustainable Development Goals (SDGs) – notably SDG1 (no poverty), SDG2 (zero hunger), SDG3 (good health & well-being), and SDG15 (life on land). While there is ambiguity regarding global drought trends over the past century (Sheffield et al., 2012; Trenberth et al., 2013; McCabe and Wolock, 2015), drought hazards will likely increase in many regions in the coming decades (Sheffield and Wood, 2008; Dai, 2011; Trenberth et al., 2013; Spinoni et al., 2017; UNDRR, 2019, Spinoni et al., 2019b). Identifying pathways towards more drought resilient societies therefore remains a global priority.

Recent severe droughts in southeastern Brazil (2014-2017), California (2011-2017), the Caribbean (2013-2016), northern China (2010-2011), Europe (2011, 2015, 2018), India (2016, 2019), the Horn of Africa (2011-2012), South Africa (2015-2016, 2018), and Viet Nam (2016), have clearly shown that the risk of negative impacts associated with droughts is not only linked to the severity, frequency, and duration of drought events, but also to the degree of exposure, susceptibility and coping capacity of a given social-ecological system. Despite this, proactive management of drought risk is still not a reality in many regions across the world. Droughts and their impacts are still mostly addressed through reactive crisis management approaches, for example, by providing relief measures (Rojas, 2018). To improve the monitoring, assessment, understanding, and ultimately proactive management of drought risk effectively, we need to acknowledge that the root causes, patterns and dynamics of

exposure and vulnerability need to be considered alongside climate variability in an integrated manner (Spinoni et al., 2019a; Hagenlocher et al., 2019).

Over the past decades, major efforts have been made to improve natural hazard risk assessments and their methodologies across scales, ranging from global risk assessments to local level assessments. At the global scale several studies have been published in recent years, focusing on the assessment of flood risk (Hirabayashi et al., 2013; Ward et al., 2013, 2014), seismic risk (Silva et al., 2018), cyclone risk (Peduzzi et al., 2012), or multi-hazard risk (e.g. Dilley et al., 2005; Peduzzi et al., 2009; Welle and Birkmann, 2015; Garschagen et al., 2016; INFORM, 2019; Koks et al., 2019; UNDRR, 2019). While major progress

has been made regarding the mapping, prediction and monitoring of drought events at the global scale (e.g. Yuan and Wood, 2013; Geng et al., 2013; Spinoni et al., 2013, 2019b; Damberg and AghaKouchak, 2014; Hao et al., 2014; Carrão et al., 2017), very few studies have assessed either exposure to drought hazards (Güneralp et al., 2015) or drought risk at the global level (Carrão et al., 2016; Dilley et al., 2005; Li et al., 2009). The study by Carrão et al. (2016) presents the first attempt to map drought risk at the global scale while considering drought hazard (based on precipitation deficits), exposure (population,

livestock, crops, water stress), and societal vulnerability (based on social, economic and infrastructural indicators). While generic drought risk assessments are useful to get an overview of the key patterns and hotspots of drought risk, it is increasingly acknowledged that drought risk assessment should be tailored to the needs of specific users, so that management plans can be developed to reduce impacts (Vogt et al., 2018; UNDRR, 2019). Impact or sector specific assessments of who (e.g. farmers) and what (e.g. crops) is at risk to what (e.g. abnormally low soil moisture, deficit in rainfall, below average streamflow), where,

and why, are needed to inform targeted drought risk reduction, resilience and adaptation strategies (IPCC, 2014). Such analyses are currently lacking. Furthermore, in their exposure analysis, Carrão et al. (2016) do not distinguish between rain-fed and irrigated agriculture, although different hazard indicators are relevant when assessing drought risk for these systems. In addition, the vulnerability analysis presented by Carrão et al. (2016) is based on a reduced set of social, economic and infrastructure-related indicators, and does not account for the role of ecosystem-related indicators as a driver of drought risk -

a gap that was recently highlighted in a systematic review of existing drought risk assessments across the globe (Hagenlocher et al., 2019). A social-ecological systems perspective, especially when assessing drought risk in the context of agricultural systems, where livelihoods depend on ecosystems and their services, can help to better understand the role of ecosystems and their services as a driver of drought risk, but also as an opportunity for drought risk reduction (Kloos and Renaud, 2016).

This paper addresses some of the above gaps by presenting, for the first time, an integrated drought risk assessment that brings together data from different sources and disciplines for rain-fed and irrigated agricultural systems considering relevant drought hazard indicators, exposure and vulnerability at the global scale. The spatial variability of drought risk on global and regional scales might help to identify leverage points for reducing impacts and properly anticipate, adapt and move towards resilient agricultural systems.

## 2 Methods

Today, it is widely acknowledged that risk associated with natural hazards, climate variability and change is a function of hazard, exposure and vulnerability (IPCC, 2014; UNDRR, 2019). Following that logic, Figure 01 shows the overall workflow of the assessment, while the subsequent sections describe in detail how drought risk for agricultural systems, including both irrigated and rain-fed systems, were assessed at the global scale.

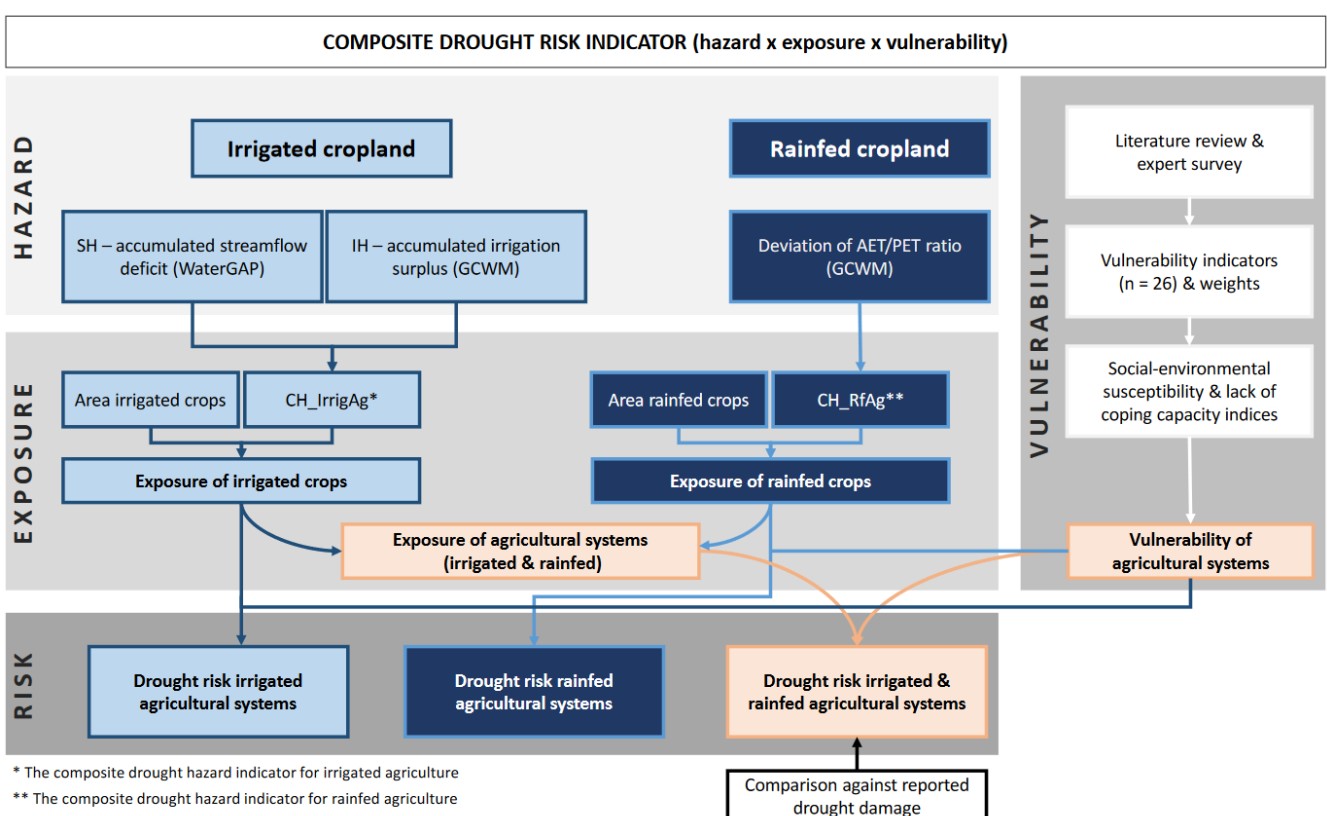

**Fig 01.** Workflow for the overall global drought risk assessment for agricultural systems (including irrigated and rain-fed systems).

The composite drought hazard indicators were calculated for irrigated and rain-fed systems separately using drought indices based on historical climate conditions (1980-2016), which resulted in integrated hazard maps for both rain-fed and irrigated agricultural systems, respectively. The different irrigated and non-irrigated crops by country were considered as the exposed element. Due to the lack of high-resolution gridded data on agricultural-dependent population at the global scale, this exposure indicator was not considered. The vulnerability component was assessed through a social-ecological systems (SES) lens, where social-ecological susceptibility and lack of coping capacity indicators were weighted by drought experts around the world.

## 2.1 Drought hazard and exposure indicators

The drought hazard indicators considered here represent the average drought hazard during the period 1980 to 2016 in each spatial unit for which it is computed. Drought hazard is defined as a deviation of the situation in a specific year or month from long-term mean conditions in the 30-year reference period from 1986 to 2015. To quantify drought hazard for such a long period, we used the global water resources and water use model WaterGAP (Müller Schmied et al., 2014) and the global crop water model GCWM (Siebert and Döll, 2010). The models simulate terrestrial hydrology (WaterGAP) and crop water use (GCWM) for daily time steps on a spatial resolution of 30 arc-minutes (WaterGAP) or 5 arc-minutes (GCWM). The most recent version WaterGAP 2.2d was forced by the WFDEI-GPCC climate data set (Weedon et al., 2014) which was developed by applying the forcing data methodology developed in the EU-project WATCH on ERA-Interim reanalysis data (Table 01). The GCWM used the CRU-TS 3.25 climate data set (Harris et al., 2014) as an input. CRU-TS 3.25 was developed by the Climate Research Unit of the University of East Anglia by interpolation of weather station observations and is provided as a time series of monthly values. Pseudo daily climate was generated by the GCWM as described in Siebert and Döll (2010). Following the definitions of the Intergovernmental Panel on Climate Change put forward in their Fifth Assessment Report (IPCC, 2014), exposure is defined as the elements located in areas that could be adversely affected by drought hazard. The distinct exposure of irrigated and rain-fed agricultural systems to drought was considered by weighting grid cell specific hazards with the harvested area of irrigated and rain-fed crops according to the Monthly Irrigated and Rain-fed Cropping Areas (MIRCA2000) dataset (Portmann et al., 2010) when aggregating grid cell specific hazards to exposure at a national scale. MIRCA2000 was also used to inform the models used in the hazard calculations about growing areas and growing periods of irrigated and rain-fed crops. The data set refers to the period centered around the year 2000; time series information is not available at the global scale. To maximize the representativeness of the land use, the reference period and evaluation period used in this study were centered around the year 2000.

**Table 01.** Hazard and exposure indicators used in the analysis and their processed data

| Risk component | Composite indicator | Indicator | Processed data |
|---|---|---|---|
| **Drought hazard** | CH_IrrigAg | Accumulated streamflow deficit | WaterGAP (1980-2016) with climate forcing WFDEI-GPCC. Streamflow monthly time series. |
| | | Accumulated irrigation surplus | GCWM (1980-2016) with climate forcing CRU TS3.25. Monthly time series of net irrigation requirements |
| | CH_RfAg | AET/PET deviation ratio | GCWM (1980-2016) with climate forcing CRU TS3.25. Annual time series of the deviation of the ratio AET / PET from the long-term (1986-2015) median of the ratio AET / PET |
| **Exposed elements** | Rainfed & irrigated | Aggregation of pixel level data to national scale | MIRCA 2000 dataset was used to compute harvested area weighted averages of the indicators |

### 2.1.1 Irrigated agricultural systems

The composite drought hazard indicator is defined as the product of mean severity and frequency of drought events. For irrigated agriculture (*CH_IrrigAg*) it combines an indicator for streamflow drought hazard (*SH*), i.e. for abnormally low streamflow in rivers, with an indicator of abnormally high irrigation water requirement (*IH*) (Fig. 01). It thus considers the deviations of both demand and supply of water from normal conditions. *SH* and *IH* are computed with a spatial resolution of 0.5° by 0.5° (55 km by 55 km at the equator). Greenland and Antarctica are excluded. As *IH* is not meaningful in grid cells without irrigation, *CH_IrrigAg* is only computed for grid cells in which irrigated crops are grown according to MIRCA2000 (Portmann et al., 2010).

*IH* was calculated by using the GCWM based on a monthly time series of net irrigation requirements from 1980 to 2016. The net irrigation requirement is the volume of water needed to ensure that the AET of irrigated crops is similar to their PET. (Fig. 01). The calculations were performed for 487,121 grid cells with a resolution of 5 arc-minutes containing irrigated crop areas and then aggregated to 26,478 grid cells with a 30 arc-minute resolution to be consistent with the resolution used by WaterGAP. *SH* was calculated by using WaterGAP based on a monthly time series of streamflow from 1980 to 2016 in 66,896 0.5° grid cells world-wide.

For both *IH* and *SH*, drought hazard per grid cell was quantified as the product of a (scaled or transformed) mean severity of all drought events during the evaluation period 1980-2016 and the frequency of drought events during this period. Drought events for *IH* and *SH* were determined independently. In the case of *IH* computation, a drought event starts as soon as the monthly irrigation requirement exceeds the irrigation requirement threshold and ends when the surplus reaches zero. In the case of *SH* computation, a drought starts if the monthly streamflow drops below the streamflow threshold and ends as soon as the deficit reaches zero. For each grid cell and each of the 12 calendar months, a drought threshold was defined as the median of the variable values in the respective calendar month during the reference period 1986-2015. To avoid spurious short droughts and drought interruptions, it was defined that 1) a drought event starts with at least two consecutive months with an IH surplus or a SH deficit and 2) one month without an IH surplus or if a SH deficit does not break the event (Spinoni et al., 2019a). The accumulated surplus (*IH*) / deficit (*SH*) during each drought event is the severity of the drought event. Mean severity is computed as the arithmetic average of the severity of all drought events during the evaluation period. As in case of *SH* the deficit and thus the severity of streamflow drought is strongly correlated with the mean annual streamflow, mean severity is therefore scaled by dividing the accumulated streamflow deficit by mean annual streamflow. In this way scaled mean streamflow drought severity is expressed as fraction of the mean annual flow volume that is on average missing during drought events. In the case of *IH*, mean severity is transformed logarithmically before computation of *IH*, as in most grid cells the

volume of irrigation water needed additionally in drought periods are relatively small (569 out of the 26,478 irrigated grid cells is lower than 100 m³; in 1,450 grids it is lower than 1,000 m³). However, there are also some grids with extremely high values (95 grids where the additional irrigation water requirement per drought event is larger than 100,000,000 m³). The logarithmic transformation accounted for the specific value distribution.

The composite drought hazard indicator for irrigated agriculture *CH_IrrigAg* was then calculated for each grid cell by combining the streamflow hazard *SH* and irrigation requirement hazard *IH*. To ensure that both indicators are weighted equally, their native values were first scaled to a range between 0 and 1 by dividing *SH* and *IH* in each grid cell by the maximum *SH* or *IH* detected globally. The frequency distribution of the *SH* values calculated that way was shifted to the left with a mean of 0.244 while the frequency distribution of *IH* was shifted to the right with a mean of 0.664. Therefore, *CH_IrrigAg* was calculated for each grid cell as:

$$CH\_IrrigAg = 0.5\big(SH/\underline{SH} + IH/\underline{IH}\big) \tag{01}$$

with *SH* being the grid cell specific streamflow hazard, *IH* being the grid cell specific irrigation requirement hazard and $\underline{SH}$ and $\underline{IH}$ being the mean of *SH* or *IH* calculated across all grid cells.

The exposure of irrigated agricultural systems to drought at national scale was derived as the harvested area weighted mean of the *CH_IrrigAg* across all grid cells belonging to the respective aggregation units.

### 2.1.2 Rain-fed agricultural systems

The composite drought hazard indicator for rain-fed agriculture (*CH_RfAg*) was quantified based on the ratio of actual crop evapotranspiration (AET - in $m^3$ $day^{-1}$) to potential crop evapotranspiration (PET in $m^3$ $day^{-1}$), calculated for the evaluation period 1980-2016 and compared to the reference period 1986-2015 (Fig. 01). PET quantifies the water requirement of the crop without water limitation while AET refers to the evapotranspiration under actual soil moisture conditions.

The GCWM was applied for 24 specific rain-fed crops and the two groups "others annual" and "others perennial" to calculate crop specific AET and PET on a daily time step. Together, the 24 crops and two crop groups cover all crop species distinguished by FAO in their database FAOSTAT. The sum of daily crop specific AET and PET was calculated for all crops and for each year in the period 1980-2016 for 927,857 grid cells containing rainfed cropland and aggregated to 37,265 grid cells with the resolution 0.5 x 0.5 degree.

The mean ratio between AET and PET ($AET/PET$) for the reference period 1986-2015 was then calculated for each grid cell. $AET/PET$ reflects long-term water limitations for the geographic unit with low values representing high aridity and high values for low aridity. *CH_RfAg* was then determined by calculating the ratio $AET/PET$ for each year from 1980-2016, and by deriving the percentile of a relative difference of 10% to the long-term mean ratio $AET/PET$ from the time series. Consequently, *CH_RfAg* reflects the probability for the occurrence of a drought year in which the ratio between total AET and total PET across all rain-fed crops is 10% lower than the long-term mean ratio $AET/PET$. We also tested other percentage thresholds (20%, 30%, 50%), but for many parts of the world we never computed reductions of the ratio AET/PET by more than 10% of the long-term mean ratio (Supplementary (S5)). Therefore, it was decided to use the 10% threshold consistently.

### 2.1.3 Integration of drought exposure of irrigated and rain-fed cropping systems

The combined drought exposure for rain-fed and irrigated cropping systems was evaluated at country level by averaging the harvested area weighted drought exposure of irrigated and rain-fed cropping systems. As described before, distinct methods were used to calculate hazard and exposure of irrigated and rain-fed systems so that a direct comparison of the exposure values is not meaningful. In addition, the frequency distributions differed considerably, with a harvested area weighted global mean of the drought exposure of 0.455 for irrigated systems and 0.189 for rain-fed systems. To ensure a more similar weight of rain-fed and irrigated drought exposure, country specific exposures were divided by the global mean, and then the integrated exposure was calculated as harvested area weighted mean:

$$Exp_{tot} = \left( \left( AH_{rf} \star Exp_{rf} \, / \, 0.189 \right) + \left( AH_{irr} \star Exp_{irr} \, / \, 0.455 \right) \right) / AH_{tot} \tag{02}$$

with $Exp_{tot}$, $Exp_{rf}$, and $Exp_{irr}$ being the exposure of the whole, rainfed and irrigated cropping systems to drought and $AH_{tot}$, $AH_{rf}$, and $AH_{irr}$ being the harvested area of all crops, rainfed crops and irrigated crops.

### 2.2 Vulnerability and risk assessment

According to the Intergovernmental Panel on Climate Change (IPCC) (2014), vulnerability is the predisposition to be adversely affected as a result of the sensitivity or susceptibility of a system and its elements to harm, coupled with a lack of coping and adaptive capacity. The assessment of drought vulnerability is complex because it depends on both biophysical and socioeconomic drivers (Naumann et al., 2014). Due to this complexity, the most common method to assess vulnerability in the context of natural hazards and climate change is using composite indicators or index-based approaches (Beccari, 2016; Sherbinin et al., 2019). Although their usefulness for policy support has also been subject to criticism (Hinkel, 2011, Beccari

2016), it is widely acknowledged that composite indicators can identify generic leverage points for reducing impacts at the regional to global scale (Sherbinin et al., 2017, 2019; UNDRR 2019).

Following the workflow to calculate composite indicators proposed by the Organisation for Economic Co-operation and Development (OECD, 2008) and Hagenlocher et al. (2018), the methodological key steps on which the vulnerability assessment is based are: 1) definition of the conceptual framework, 2) identification of valid indicators, 3) data acquisition and pre-processing, 4) analysis and imputation of missing data, 5) detection and treatment of outliers, 6) assessment of multicollinearities, 7) normalization, 8) weighted aggregation, and 9) visualization.

An initial set of vulnerability indicators for agricultural systems was identified based on a recent review of existing drought risk assessments (Hagenlocher et al., 2019). In total 64 vulnerability indicators, including social, economic, physical, farming practices, environmental, governance, crime and conflict factors, were selected and classified by social-ecological susceptibility (*SOC_SUS, ENV_SUS*), lack of coping capacity (*COP*) and lack of adaptive capacity (*AC*) following the risk

framework of the IPCC (IPCC, 2014). Indicator weights, which express the relevance of the identified indicators for characterizing and assessing the vulnerability of agricultural systems to droughts, were identified through a global survey of relevant experts ($n = 78$) around the world; the majority of whom have worked in academia and for governmental organizations with more than five years of work experience (Meza et al., 2019). In total, 46 of the 64 indicators were considered relevant by the experts, comprising susceptibility, coping and adaptive capacity indicators. However, since adaptive capacity is only

relevant when assessing future risk scenarios and less relevant to current risk, indicators related to adaptive capacity and indicators that could be measured with the same data source due to their similarity in what they represent were removed. For instance *Agriculture (% of GDP)* and *Dependency on agriculture for livelihood (%)* were averaged into one income indicator and the variables *GDP per capita, PPP* and *Population below the national poverty line (%)* both refer to poverty, and therefore were also averaged to a combined indicator. This resulted in a set of 26 indicators as part of the vulnerability assessment (Table

02).

Following data acquisition, the data were pre-processed by transforming absolute to relative values and standardized when necessary (e.g. travel time to cities ≤30 min (population), divided by the total population). Descriptive statistics were used to evaluate the degree of missing data. The imputation of missing values was done with data from previous years and using

secondary sources following Naumann et al. (2014) in cases where the *r* value lay between -1.0 to -0.9 or 1.0 to 0.9 using a Spearman correlation matrix and scatter diagram for visual interpretation. Following suggestions by Roth et al. (1999), Peng et al. (2006) and Enders (2003), listwise and pairwise deletion thresholds were selected when >30% of data were missing on a country level and when > 20% of data were missing on the indicator level. After the deletion, 168 countries and 26 indicators were considered for the final analysis. To detect potential outliers, scatter plots and box plots for each indicator were created.

Potential outliers were further examined using triangulation with other sources and past years. On this basis, outliers were

identified in only one indicator (i.e. fertilizer consumption (kg/ha of arable land)) and treated using winsorization following Field (2013). Multicollinearities were identified using a Spearman correlation matrix for the different vulnerability components (social susceptibility, environmental susceptibility and lack of coping capacity). Following the rule proposed by Hinkle et al. (2003), any values higher than $r > 0.9$ or smaller than $r < -0.9$ were considered very highly correlated. The correlation was considered only if it was significant at the 0.05 level (2-tailed). Two indicators for the lack of coping capacity component and two from social-susceptibility (e.g. *healthy life expectancy at birth (years)*, and *disability-adjusted life*) showed high and significant correlations. However, no indicators were excluded on this basis, due to the difference in concepts they represented and their relevance at global level. In order to render the indicators comparable, the final selected indicators were normalized to a range from 0 to 1 using min-max normalization (Naumann et al., 2014; Carrão et al., 2016):

$$Z_i = X_i - X_{min} / X_{max} - X_{min} \tag{03}$$

where $Z_i$ is the normalized score for each indicator score $X_i$. For variables with negative cardinality to the overall vulnerability the normalization was defined as:

$$Z_i = 1 - (X_i - X_{min} / X_{max} - X_{min}) \tag{04}$$

Finally, the normalized indicator scores were aggregated into vulnerability components (*SOC_SUS, ENV_SUS, COP*) using weighted arithmetic aggregation based on (using the example of *SOC_SUS*):

$$SOC\_SUS = \sum W_i Z_i \tag{05}$$

where $W_i$ are the weights for each normalized dataset, and $Z_i$ are the weights as obtained from the global expert survey. Thereby, weights were normalized to add up to 1. The final indicators and their respective weights are listed in Table 02. The vulnerability components of social-ecological susceptibility (*SE_SUS*) were combined using an average, which was then combined with lack of coping capacity (*COP*) to obtain a final vulnerability index (*VI*) score:

$$VI = V(SE\_SUS) + V(COP) / 2 \tag{06}$$

**Table 02.** Vulnerability indicators used in the analysis and their related expert-weights*.

| Indicator | Data source | Weight* |
|---|---|---|
| **Social susceptibility (SOC_SUS)** | | |
| Share of GDP from agr., forestry and fishing in US$ (%) | FAO (2016) | 0.96 |
| Rural population (% of total population) | World Bank (2011-2017) | 0.85 |
| Prevalence of undernourishment (% of population) | World Bank (2015) | 0.82 |

| | | |
|---|---|---|
| Literacy rate, adult total (% of people ages 15 and above) | World Bank (2015) | 0.80 |
| Prevalence of conflict/insecurity (Crime and Theft, Index (0-30)) | World Bank (2017) | 0.76 |
| Proportion of population living below the national poverty line (%) | SDG indicators (2015-2017) | 0.75 |
| Access to improved water sources (% of total population with access) | World Bank/FAO (2015) | 0.66 |
| DALYs (Disability-Adjusted Life Years)(DALYs per 100,000, Rate) | GBD (2016) | 0.65 |
| GINI index | World Bank (2017) | 0.64 |
| Insecticides and pesticides used (ton/ha) | FAO (2016) | 0.63 |
| Gender Inequality Index | UNDP (2018) | 0.62 |
| Electricity production from hydroelectric sources (% of total) | World Bank (2015) | 0.62 |
| Unemployment, total (% of total labor force) (national estimate) | World Bank (2017) | 0.60 |
| Dependency ratio (Population ages 15-64 (% of total population)) | World Bank (2011-2016) | 0.60 |
| Population using at least basic sanitation services (%) | WHO (2015) | 0.60 |
| Healthy life expectancy (HALE) at birth (years) | WHO (2014) | 0.56 |
| **Ecological susceptibility (ECO_SUS)** | | |
| Average land degradation in GLASOD erosion degree | FAO (1991) | 0.92 |
| Fertilizer consumption (kilograms per hectare of arable land) | World Bank (2015) | 0.74 |
| Average soil erosion | FAO (1991) | 0.72 |
| Terrestrial and marine protected areas (% of total territorial area) | World Bank (2016-2017) | 0.63 |
| **Lack of coping capacity (COP)** | | |
| Saved any money in the past year (% age 15+) | Global FINDEX (2014-2017) | 0.87 |
| Government Effectiveness: Percentile Rank | World Bank (2017) | 0.85 |
| Total dam storage capacity per capita. Unit: m3/inhab | FAO Aquastat (2017) | 0.82 |
| Total renewable water resources per capita (m3/inhab/year) | FAO (2014) | 0.76 |
| Corruption Perception Index (CPI) | Transparency International (2017) | 0.68 |
| Travel time to cities ≤30 min (population) (%) | JRC (2015) | 0.65 |

 *derived from a global expert survey (Meza et al., 2019)*

The final drought risk index (DRI) (Fig. 01) was calculated by multiplying the indices for drought hazard/exposure and vulnerability. At pixel level, the presence of hazard and vulnerability point to a certain drought risk, independent of how much crop area is contained in the specific pixel. At aggregated level, the different crop areas in the specific pixels must be considered; therefore exposure was calculated as harvested area weighted mean of the pixel level hazard and then multiplied with vulnerability to calculate drought risk at country level.

The total drought risk score for irrigated and rain-fed systems combined ($DRI_{tot}$) is derived by multiplying the exposure of the whole cropping system $Exp_{tot}$ (Equation 02) with the vulnerability index $VI$.

**2.3 Comparison against drought impact data**

The outcomes of the risk assessment for irrigated and rain-fed systems combined ($DRI_{tot}$) were compared against impact data from the international Emergency Events Database (EM-DAT) of the Centre for Research on the Epidemiology of Disasters (CRED) using visual correlation (Fig. 06). EM-DAT systematically collects reports of drought events and drought impacts from various sources, including UN agencies, NGOs, insurance companies, research institutes and press agencies. Here, the number of drought events within the period 1980-2016 was used as an input for the comparison. Thereby, a drought event is registered in EM-DAT when at least one of the following criteria applies: 10 or more people dead; 100 or more people affected; declaration of a state of emergency or a call for international assistance.

**3 Results**

This section presents the results of the global drought risk assessment for agricultural systems (irrigated and rain-fed) at pixel level (Fig. 02 and 03) and for the total risk of both systems combined at national resolution (Fig. 04). The patterns colored dark red show high levels of the different risk components, while the dark blue colors reflects low scores of the different risk components.

**3.1 Drought risk for irrigated agricultural systems**

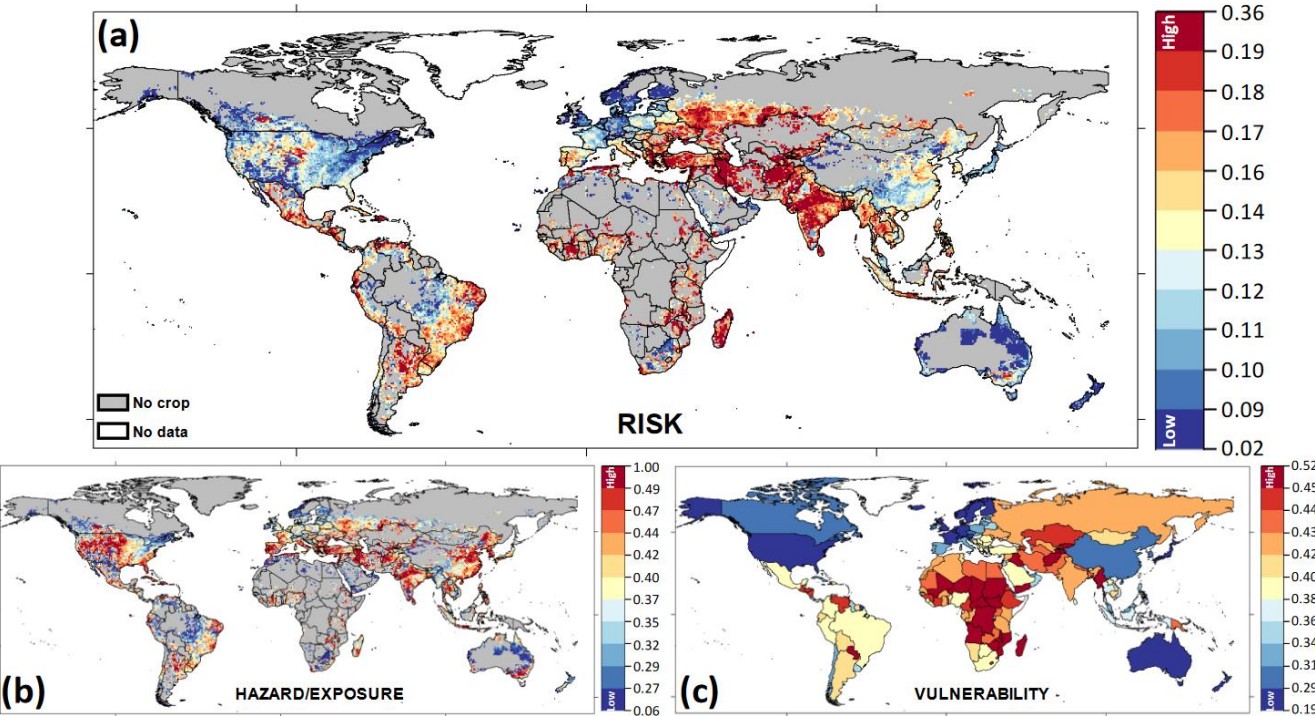

**Fig 02**. Drought risk (a), hazard/exposure (b) and vulnerability (c) for irrigated agricultural systems. The legends were defined by assigning the median of the value distribution to the yellow color in the center, the 90th percentile to the deepest red color, the 10th percentile to the deepest blue color, and by determining the class ranges of the other colors by linear interpolation. Risk was directly calculated by multiplying hazard with vulnerability (pixel-level analysis).

The drought risk for irrigated agricultural systems varies significantly among continents and countries. Especially large countries such as USA, Brazil, China and Australia show a high variation at country level, due to varying climatic conditions. Drought hazard/exposure was highest in regions with a high density of irrigated land and high irrigation water requirements such as the western part of USA, central Asia, northern India, northern China and southern Australia. Vulnerability was high particularly in sub-Saharan Africa but also in some countries in central Asia and the Middle East region and low in general for industrialized and high income countries. The combination of hazard and vulnerability to risk resulted in highest values for large parts of west, central and south Asia, eastern Africa and the eastern part of Brazil. Low risk areas include western Europe, USA, Australia and most parts of China (Fig. 02).

## 3.2 Drought risk for rain-fed agricultural systems

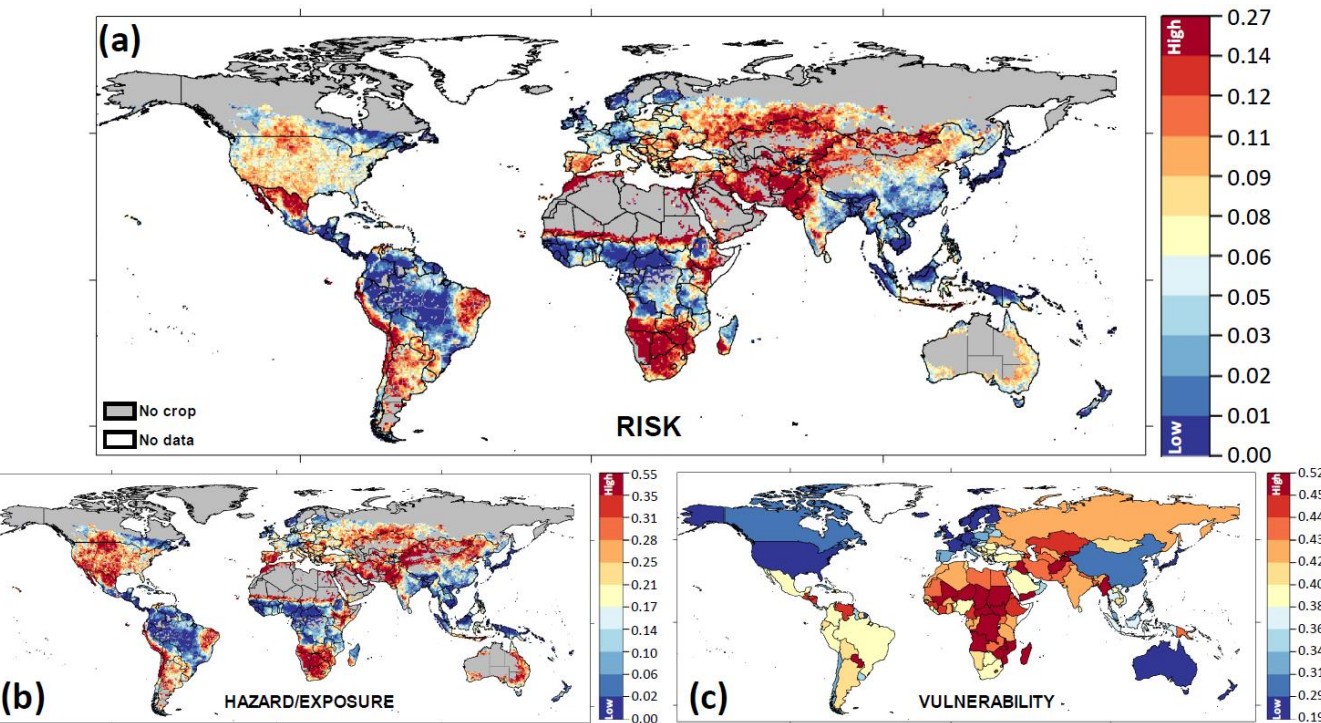

**Fig 03**. Drought risk (a), hazard/exposure (b) and vulnerability (c) for rain-fed agricultural systems. The legends were defined by assigning the median of the value distribution to the yellow color in the center, the 90th percentile to the deepest red color, the 10th percentile to the deepest blue color and by determining the class ranges of the other colors by linear interpolation. Risk was calculated by multiplying hazard/exposure with vulnerability (pixel level analysis).

High levels of risk (dark yellow to red color scheme) for rain-fed agricultural systems are observed in southern Africa, southeastern Europe, northern Mexico, northeast Brazil, at the western coast of South America, southern Russia and western Asia. The vulnerability to drought highlights the relevance to increase the coping capacity of the countries in order to reduce their overall drought risk. For instance, Australia, despite being highly exposed to drought hazard, has low socio-ecological susceptibility and high enough coping capacities to considerably reduce the overall drought risk.

**3.3 Drought risk for agricultural systems (irrigated and rain-fed combined)**

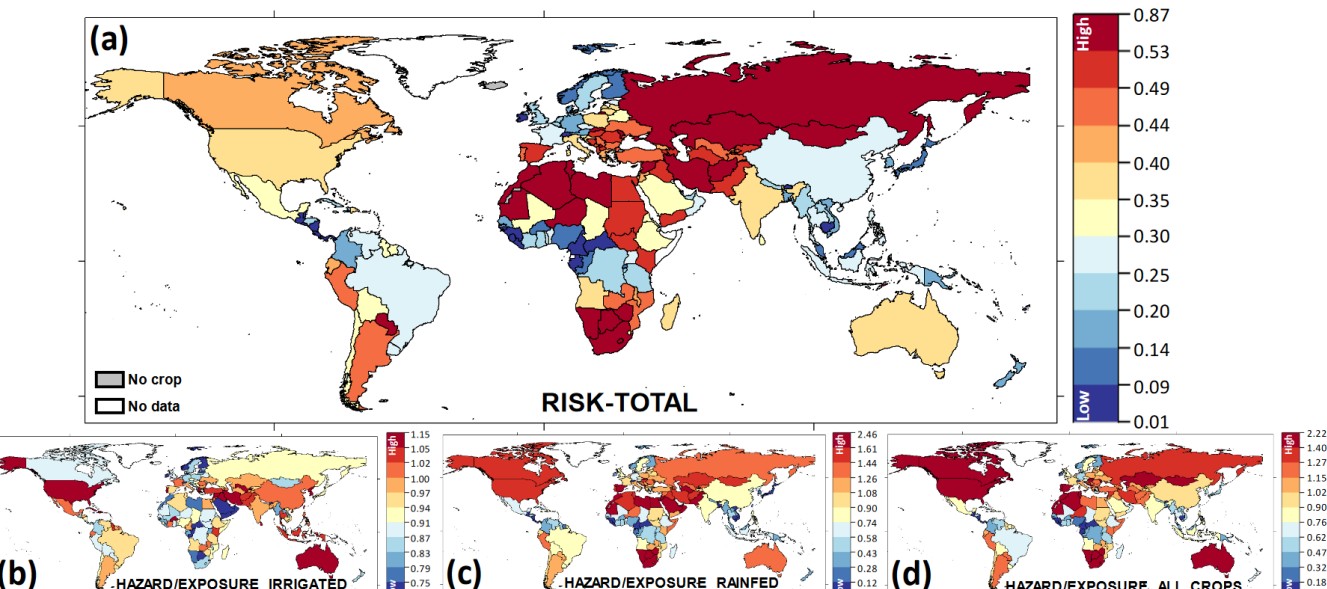

**Fig 04**. Drought risk (a), hazard/exposure of irrigated (b), rain-fed (c), and the whole crop production sector (d). The legends were defined by assigning the median of the value distribution to the yellow color in the center, the 90th percentile to the deepest red color, the 10th percentile to the deepest blue color, and by determining the class ranges of the other colors by linear interpolation. Risk was calculated by multiplying hazard/exposure with vulnerability shown in Fig. 02c and 03c.

The hazard/exposure maps shown in Figure 04 are slightly different to the ones shown in Figures 02 and 03 due to the aggregation at country level. The analysis shows that regions with low hazard/exposure of rain-fed and irrigated crops to drought tend to be tropical and subarctic regions following the Köppen-Geiger climate classification (1980-2016) (Beck et al., 2018). There are significant regional differences when comparing irrigated and rain-fed drought hazard/exposure. For instance, the northern part of Latin America and Central Africa have low hazard/exposure levels, given the humid climate conditions

resulting in a low total risk, even though those regions are characterized by high vulnerability levels. Southern Africa, however, has a high amount of drought-exposed rain-fed crops, but a lower vulnerability compared to other African countries. Despite this, risk scores in that region are very high. Very high drought hazard/exposure and vulnerability levels can be found in the Middle East and Northern Africa.

Although the drought hazard was computed differently for the different agricultural systems, the countries with high risk of drought to both farming systems are Botswana, Namibia and Zimbabwe (Fig. 02 and 03), these countries share the same relevant indicators that define their high vulnerability: high soil and land degradation rate, low literacy rate, and low total

renewable water (Supplementary (S3)). Table 03 shows the top and bottom ten countries with the highest/lowest total drought risk ($DRI_{tot}$) as well as their hazard/exposure and vulnerability scores.

**Table 03.** Rank of countries with the highest and lowest risk of drought for combined agricultural systems (rain-fed and irrigated)

| Country | Drought risk (countries rank) | Risk score total | Hazard/Exposure | | | Vulnerability score |
|---|---|---|---|---|---|---|
| | | | Haz/Exp irrigated | Haz/Exp rain-fed | Haz/Exp total | |
| Zimbabwe | 1 | 0.871 | 0.967 | 1.885 | 1.804 | 0.483 |
| Namibia | 2 | 0.846 | 0.769 | 2.122 | 2.061 | 0.411 |
| Botswana | 3 | 0.811 | 0.466 | 2.095 | 2.076 | 0.391 |
| Morocco | 4 | 0.786 | 0.774 | 2.172 | 1.873 | 0.419 |
| Kosovo | 5 | 0.728 | 0.936 | 1.871 | 1.854 | 0.393 |
| East Timor | 6 | 0.701 | 0.971 | 1.882 | 1.854 | 0.378 |
| Mauritania | 7 | 0.692 | 0.886 | 1.670 | 1.580 | 0.438 |
| Lesotho | 8 | 0.692 | 0.840 | 1.562 | 1.556 | 0.445 |
| Kazakhstan | 9 | 0.670 | 0.974 | 1.573 | 1.499 | 0.447 |
| Algeria | 10 | 0.636 | 0.969 | 1.595 | 1.492 | 0.426 |
| Guatemala | 158 | 0.039 | 0.857 | 0.026 | 0.087 | 0.446 |
| Gambia | 159 | 0.037 | 0.760 | 0.093 | 0.094 | 0.394 |
| Belize | 160 | 0.035 | 0.943 | 0.079 | 0.093 | 0.375 |
| Sierra Leone | 161 | 0.023 | 0.934 | 0.005 | 0.057 | 0.402 |
| Brunei | 162 | 0.020 | 0.741 | 0.000 | 0.077 | 0.254 |
| Guinea | 163 | 0.019 | 0.822 | 0.033 | 0.042 | 0.452 |
| Switzerland | 164 | 0.017 | 0.695 | 0.046 | 0.068 | 0.247 |
| Guinea-Bissau | 165 | 0.017 | 0.723 | 0.026 | 0.042 | 0.401 |
| Fiji Islands | 166 | 0.011 | 0.833 | 0.017 | 0.033 | 0.329 |
| Central African Republic | 167 | 0.008 | 0.646 | 0.016 | 0.016 | 0.505 |

Seven out of the ten countries with the highest overall drought risk are located on the African continent. However, Kosovo, East Timor and Kazakhstan also possess high risk levels (Table 03). Zimbabwe ranks as the country with the highest drought risk mainly due to its high exposure combined with its high vulnerability (S1).

In general, the countries that present higher drought risk have a high amount of exposed crops. Vulnerability varies among them, with Zimbabwe being the country with the highest vulnerability. The lack of coping capacity and social-ecological susceptibility were determinant factors for countries like Botswana and Zimbabwe (S1). There were cases where countries such as Namibia presented high socio-ecological susceptibility in contrast with high coping capacity, reducing its overall

vulnerability. The drought risk in countries such as Lesotho and Mauritania that have in contrast limited coping capacities, is notably higher (S1). The analysis also reveals that, although risk is currently close to zero in several countries (e.g. Fiji Islands, C.A.R., Guinea-Bissau etc.), this could rapidly change once these countries are affected by droughts given their very high vulnerability.

The comparison of the drought risks of rain-fed and irrigated cropping systems (Fig. 05) shows that several countries such as Zimbabwe, Iraq and Algeria are exposed to high risk for both cropping systems. These countries are frequently hit by drought and similarly have a high vulnerability to drought (Fig. 02 and 03). In contrast, countries such as Switzerland, Finland and New Zealand are characterized by low drought hazard/exposure for irrigated and rain-fed systems and low vulnerability to drought (Fig. 02 and 03). In countries such as Botswana, Oman and the United Arab Emirates, drought risk is high for rain-

fed cropping systems but low for irrigated cropping systems (Fig. 05). These countries are defined by arid climate conditions exposing rain-fed crops to high risk while the drought risk for irrigated cropping systems is low because of relatively low interannual variability in climatic conditions resulting in low variability of irrigation water requirement and streamflow, their risk is also determined by their different vulnerability dynamics (e.g. hydroelectric sources, retain renewable water). In contrast, drought risk for irrigated cropping systems is high and drought risk of rain-fed cropping systems is small in countries

such as Burkina Faso, Madagascar and Cote D'Ivoire (Fig. 05). In these three countries, there is a big variability in climatic conditions with irrigated crops being cultivated in the more arid parts of the country and rain-fed crops in more humid parts. In addition, aquatic crops with a high water demand such as rice and sugarcane, are the most commonly cultivated irrigated crops in these countries (Frenken, 2005).

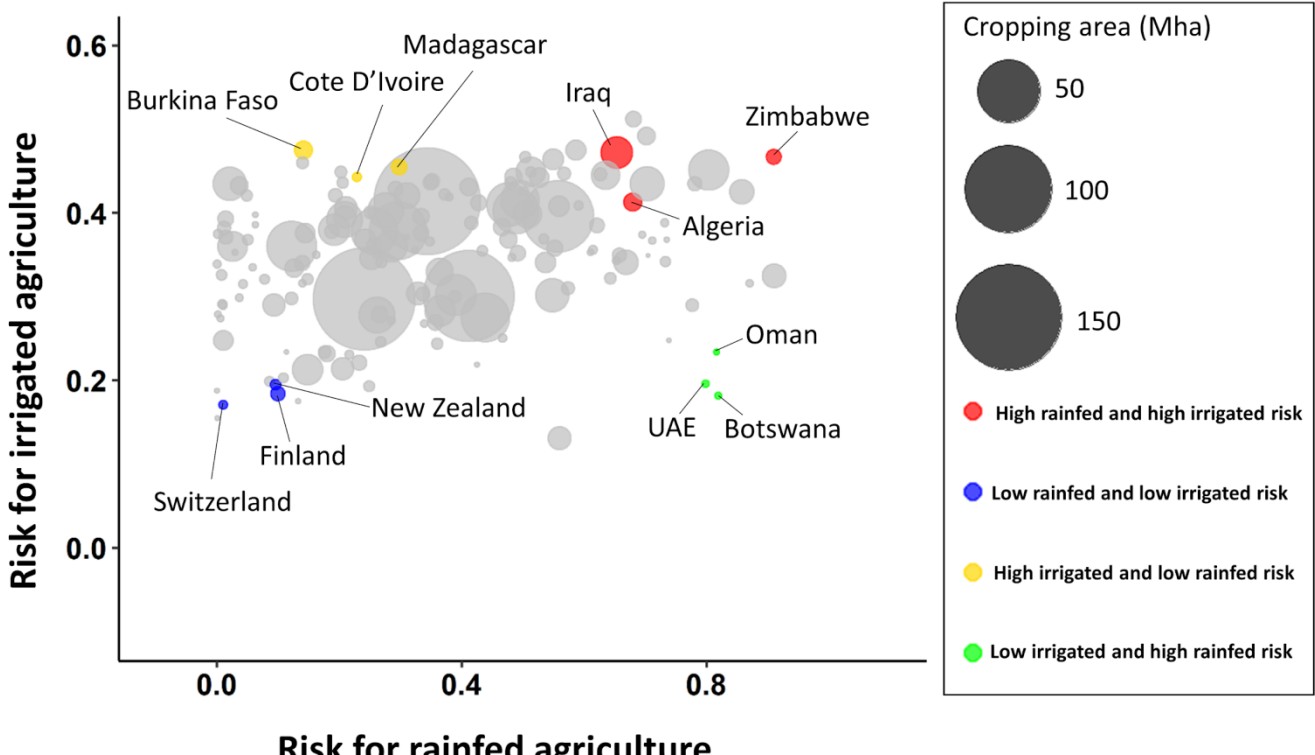

**Fig. 05** Country profiles contrasting the drought risk of irrigated and rain-fed agricultural systems. The size of the bubbles indicates the crop growing area (sum of rain-fed and irrigated areas per country in Million ha).

### 3.4 Comparison

The comparison of drought risk ($DRI_{tot}$) with drought events registered in EM-DAT shows good agreement in many countries. For countries which have low drought risk such as the countries in tropical Africa, northern and western Europe or countries in the northern part of South America, there is either none or just one drought registered in EM-DAT (Figure 06a, 06b). There is also good agreement for countries in southern Africa and some countries in the African transition zone with very high drought risk and many registered drought events and for countries with intermediate drought risk such as Canada, Australia or Italy. However, some disagreement between calculated risk and the number of reported drought events is acknowledged. For instance, Brazil is not showing high agreement between EM-DAT and the country risk level, even though the eastern part of the country presents a high risk for irrigated and rainfed systems (Fig. 2 and 3), total drought risk level is affected by the other regions with lower risk in the country. The same occurs in other large countries such as USA, Russia, China and India, the calculated drought risk is low or intermediate although a large number of drought events have been registered in EM-DAT. The reason for this disagreement is that the risk shown in Figure 06a is representative for the whole country while drought

events which only have local or regional impacts are also registered in EM-DAT (see Sect. 2.3). For all these big countries, we detected considerable spatial heterogeneity with regard to drought risk where regions with high drought risk such as the central part of USA, northeast Brazil, northern China and northwest India are complemented with other regions of low drought risk (Fig. 06a). Therefore, the high number of registered drought events in EM-DAT is corroborated by the presence of high regional drought risk (Fig. 02 and 03).

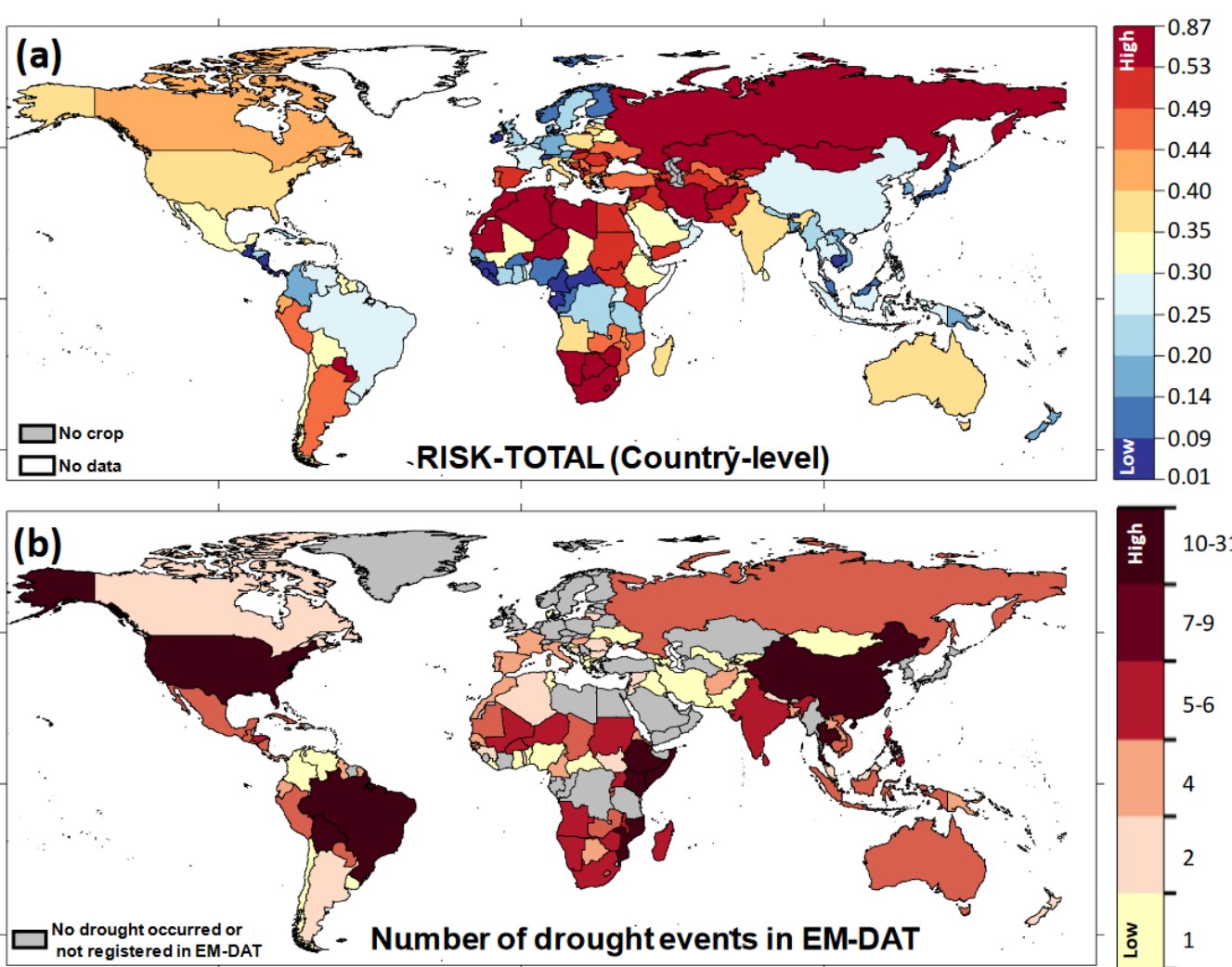

**Fig 06.** Comparison of total risk against drought impact data

## 4 Discussion

The present study performs, for the first time, a separate global drought risk analysis for irrigated and rain-fed cropping systems, including regions that indicate a high vulnerability to droughts and are particularly exposed. In previous assessments, the share of irrigated cropland was either ignored or considered as a vulnerability indicator (Carrão et al., 2016). The drought hazard analysis is based on three indicators: streamflow drought hazard ($SH$), abnormally high irrigation water requirement ($IH$), and a composite drought hazard indicator for rain-fed agriculture ($CH\_RfAg$), which quantify drought as a deviation from normal conditions consistent with common definitions. In agreement with the results for drought hazard obtained by Carrão et al. (2016), the largest drought hazard is obtained for arid and semi-arid regions such as northern and southern Africa, northern Mexico, along the coastline of Peru and Chile, the Arabian Peninsula and Mongolia for rain-fed systems, Italy, Turkey and Western Mexico for irrigated systems, and the western USA, northeast Brazil, western Argentina, central Asia, Middle East countries, western India, northern China and southern Australia for both irrigated and rain-fed systems. In contrast, previous studies based on standardized indices such as the SPI have detected the highest drought hazard mainly in humid regions such as central Europe, southeast Asia, southern Brazil and tropical Africa (Geng et al., 2015). The reason for this difference could be that deviations from normal conditions should not be treated similarly for arid and humid regions as not every precipitation or streamflow deficit in humid regions will automatically become a hazard for cropping systems. In fact, in humid regions, crops often perform better in relatively dry years (Holzkamper et al., 2015). We account for these effects by normalizing streamflow deficits with long-term mean annual river discharge ($SH$) or by calculating the probability of reductions in the AET/PET ratio of rain-fed crops in relative terms ($CH\_RfAg$).

In the present study, the rain-fed hazard is computed as the probability of a 10% decline in the AET/PET ratio compared to long-term mean conditions, whereas the irrigated drought hazard represents the combination of severity and frequency values derived from streamflow or irrigation water requirement (see Sect. 2). While the methodology reflects well the common understanding of the factors most influential for drought hazard in the two cropping systems, a direct numerical comparison of the calculated hazard for rain-fed and irrigated systems is not meaningful. The hazards and exposure calculated in this study should be used to rank or compare countries within the rain-fed or irrigated domain but not in between. The reasoning for the calculation of the total exposure and risk in this study was less to support comparisons across countries but to account for the different extend of irrigated and rainfed systems within the specific countries. There are countries in which crop production is completely rainfed and countries in which all crops are irrigated so that only the risk for the rainfed or irrigated systems are relevant. Except from these extremes, crop production in most countries is either predominantly irrigated or predominantly rainfed. We account for this by calculating total crop exposure to drought (Fig. 04d) as the harvested area weighted mean of the exposures of irrigated crops (Fig. 04b) and of the rainfed crops (Fig. 04c) Our attempt to calculate hazard, exposure and risk for the whole crop production sector by assigning a similar weight to the hazard-exposures for rain-fed and irrigated systems must be viewed critically and results should be taken with care. A potential way to derive specific weights for rain-

fed and irrigated exposure could be validating calculated hazard and exposure, but also vulnerability and risk, with information about drought impacts separately, for both irrigated and rain-fed systems. Lack of data for drought impacts distinguishing rain-fed and irrigated systems was the main reason why this approach was not implemented for the current study.

The calculation of the drought hazard of irrigated cropping systems in this study is based on the two components streamflow hazard ($SH$) and irrigation requirement hazard ($IH$) reflecting water supply ($SH$) and water demand ($IH$) of irrigated systems. Therefore we do not consider specifically in our approach the availability and use of groundwater resources for irrigation. It is well agreed that dynamics in streamflow are usually larger than dynamics in groundwater storage, so that groundwater is used by many farmers to substitute temporary deficits in surface water supply for irrigation systems. In general, access to groundwater should therefore be considered to reduce drought hazard and vulnerability of irrigated cropping systems. Consideration of groundwater resources would, however, require dynamic quantification of groundwater storage and groundwater levels, which is challenging for global scale analyses and not possible with the models applied in this study. In addition, more conceptual work is needed to decide which degree of temporal variability in groundwater levels constitutes a hazard and how to treat long-term depletion of groundwater resources (negative trends) in drought risk studies.

The multi-dimensional nature of vulnerability of agricultural systems is represented by a set of 26 expert-weighted indicators. One of the major limitations of this data driven approach is the spatial detail information for computing the model, however, at a global level it is not feasible to get a harmonized dataset of all the proxy variables, but some caution must be advised when zooming in at the subnational level (Naumann et al. 2018). When interpreting the results, it is necessary to consider that some highly correlated indicators were maintained in the analysis as they present different drivers of vulnerability and hence different entry points for vulnerability reduction. The selected indicators comprise social, economic, environmental, physical, and governance-related factors contributing to social-ecological susceptibility and the lack of coping capacity. In doing so, the present study goes beyond existing global drought risk assessments (Carrão et al., 2016) which are based on equal weights and do not consider relevant environmental vulnerability indicators as a driver of drought risk. The latter, however, is relevant when assessing drought risk for agricultural systems, where factors such as land degradation or soil erosion are shown to exacerbate drought risk (Hagenlocher et al., 2019). In future assessments an alternative to the expert-based weighting of vulnerability indicators chosen here could be the use of statistical approaches (e.g. Principal Component Analysis) to identify relevant indicators. However, given the high number of experts who participated in the weighting exercise (n = 78) the expert-based approach seems more suitable to identify relevant indicators as compared to an approach that builds on statistical significance only. Further, Hagenlocher et al. (2013) evaluated the outcomes of PCA-based and expert-based indicator choice on a composite vulnerability index, and did not find major differences.

The findings of the drought risk assessment presented here correspond to a certain degree to the findings of Carrão et al. (2016). Although the focus of the current paper is more explicitly on agriculture, both studies present methodological similarities. In Carrão et al. (2016) the percentage crop land per grid-cell is one factor in the exposure analysis and the percentage irrigated agricultural land is one of the vulnerability factors. Although Carrão et al. (2016) include other factors such as population density, livestock density and baseline water stress in the analysis, the results give a high weight to the risk for agriculture. In both studies the regions less affected by droughts correspond to the regions with low or no exposure for agriculture and population (e.g. deserts and tropical forests). This is mainly the case in Amazonia and Central Africa. Also, similarities between areas of high levels of risk are evident, including southern and eastern Europe, the Eurasian steppe, northern Africa and the Middle East, northeastern Brazil and south eastern South America.

Similarities are also found for the risk of irrigated agricultural systems. Examples are irrigated croplands in India, the United States and Australia. Differences in the overall patterns are due to the separation of irrigated and non-irrigated agriculture in the current study and the aggregated exposure information in Carrão et al. (2016). In an updated version of the risk map from Carrão et al. (2016), using a higher resolution population database and grid level exposure information, as shown in Vogt et al. (2018, Figure 7) similarities are even more evident.

However, the present study includes a spatially explicit model of AET for the main crop types of two different agricultural systems (irrigated and rain-fed agriculture), and includes a specialized vulnerability index for this sector according to expert judgment. These differences have revealed the importance to focus more clearly on distinct impacts (e.g. on irrigated vs rainfed systems) when conducting drought risk assessments, even within the same sector. For instance, irrigated agricultural systems in Latin America are highly exposed to droughts, whereas the probability of droughts occurring in rain-fed agricultural systems in that region is comparably low.

Despite these advancements, the presented analysis does have limitations. First, due to the lack of up to date land use data on irrigated vs. rain-fed agriculture at global scale, the exposure analysis is based on MIRCA data from the year 2000 (Portmann et al., 2010). Given that cropping systems are subject to change, this adds uncertainty to the results. Second, data used for the vulnerability analysis stems from different sources which makes it difficult to evaluate the inherent uncertainties in the data. Third, the data is not consistently available for all countries for the same years (Table 02). Fourth, the vulnerability analysis is based on nation-state resolution data, which does not allow for mapping spatial variability in vulnerability at the sub-national level. Fifth, applying expert opinions to weight drought vulnerability indicators according to their relevance brings subjectivity to the assessment, which necessitates a strong network of relevant experts. Sixth, preventive/adaptive planning requires going beyond evaluating drivers of risk and mapping current patterns of risk. Future scenarios of drought risk, considering both

changing environmental and climate conditions as well as possible future socio-economic development pathways, are needed in order to anticipate future challenges.

Future research should address these challenges by also investigating sub-national patterns in vulnerability, and developing future drought risk scenarios in all dimensions of drought hazards, exposure, and vulnerability. In addition, attempts to investigate changes and trends in drought risk and risk components are highly needed to better understand trajectories of drought risk in different countries and for the whole world. Further, inherent uncertainties, as well as the sensitivity of the risk assessment outcomes towards changes in the input parameters (e.g. indicator choice and weighting), should be investigated

and validated statistically. This gap has also been highlighted in a recent review of climate vulnerability assessments (Sherbinin et al., 2019) in general, as well as in a recent review of drought risk assessments (Hagenlocher et al., 2019) in particular.

The comparison conducted in this study, has shown that there is limited data available on agricultural losses and impacts caused by droughts at the global level. Furthermore, impacts are not always direct, as droughts can have cascading indirect

impacts (Freire-Gonzales et al., 2017; Van Lanen et al., 2017) which are difficult to assess. In addition, for countries where we find high drought risk (e.g., Mongolia, Iran, Kazakhstan and the countries in southeast Europe), no or very few drought events are registered in EM-DAT. The reason for this mismatch could be that drought events in these countries were not registered in EM-DAT. For example, in Romania, EM-DAT reports two drought events while according to other reports, twelve years between 1980 and 2012 were classified as drought years with 48% of the agricultural land affected (Lupu et al.,

2010; Mateescu et al., 2013). On top of this, in Iran, EM-DAT reports one drought event while other sources recounted several droughts during 1980-2005, with the most extreme drought lasting for four years from 1999 to 2002 (Javanmard et al., 2017; Zoljoodi and Didevarasl, 2013). These examples suggest that it cannot be concluded from missing drought records in EM-DAT that specific countries were not affected by drought. Once improved and reliable impact data is available at the global scale, future research should also focus on the statistical validation of drought risk assessments with drought events and impact

data. Ongoing efforts of countries to report their losses and impacts due to natural hazards (e.g. as part of the Sendai Monitoring) are considered as a first important step towards that direction.

Lastly, while this study presents the first attempt to assess drought risk for agricultural systems, more work is needed to analyze drought risk for other sectors, such as public water supply, tourism, energy production, and water-borne transport, among

others.

## 5 Conclusions

This paper presents, for the first time, a global-scale drought risk assessment for both irrigated and rain-fed agricultural systems from a social-ecological perspective by integrating drought indicators for hazard, exposure, and vulnerability. It goes beyond

previous studies by including a separated and spatially explicit analysis of the drought hazard and exposure for irrigated and rain-fed agricultural systems, as well as an empirically-based weighting of vulnerability indicators. The latter being based on the judgment of drought experts around the globe. The presented methodology can serve as a framework for the analysis of other affected sectors, such as water or energy. Findings from this study underscore the relevance of analyzing drought risk from a holistic perspective (i.e., including the sector-specific hazard, exposure and vulnerability) and based on a spatially explicit approach. By providing information on high risk areas and underlying drivers, this approach helps to identify priority regions as well as entry points for targeted drought risk reduction and adaptation options. While this first attempt provides valuable information at the global level, improvements could be achieved with the availability of more spatially explicit vulnerability information (i.e. at sub-national levels) and the availability of standardized drought impact information that can serve for a quantitative validation of risk levels.

## 6 Acknowledgments

The research is part of the project GlobeDrought (grant no. 02WGR1457A-F) funded by the German Federal Ministry of Education and Research (BMBF) through its Global Resource Water (GRoW) funding initiative. The authors would like to thank the 78 experts for their participation in the global expert survey, the two reviewers for their valuable comments, and Harrhy James for proofreading of the revised manuscript.

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
