# Peer review of "Global-scale drought risk assessment for agricultural systems"

_Natural Hazards and Earth System Sciences, 2019_

## Referee Comment (RC1) · Anonymous Referee #1 · 12 Sep 2019

The authors present a relevant and interesting manuscript, where they have studied and mapped composite drought risk at the global scale. For assessing agricultural drought risk, they have separated drought hazard/exposure in irrigated and rainfed cropping systems, and combined these hazard indicators with socio-ecological vulnerability. Finally, they have compared the obtained drought risk metric, with reported drought hazard events from EM-Dat. In general, I like and agree with the approach described in the study. My notes about the study are written below.

1. I very much agree with looking into drought hazard for irrigated and rainfed cropping systems separately. However, the way these hazard indicators are combined, is potentially misleading. The hazard indicators are combined in a way that equalizes the weight of drought hazards in irrigated and rainfed cropping systems. However, as

[Figure]

irrigated systems are, in general, more resilient to drought (irrigated systems can mitigate drought impacts by irrigation while rainfed systems cannot), equalizing the hazard associated with rainfed and irrigated systems, does not seem sensible. Further, if I understand correctly, based on the analyses, drought is more frequent in irrigated cropping systems compared to rainfed systems, which is not something that would be initially expected (Page 7, Lines 193-194). To make the methods comparable across rainfed and irrigated cropping systems, the authors could potentially define droughts for rainfed systems as for irrigated systems, but without the option to compensate the demand deficit by irrigation.

2. The vulnerability assessment includes a high number of indicators. Although, the authors have excluded variables that have >0.9 correlation, many of the indicators are still most likely highly correlated. Considering the method used for calculating the vulnerability metric, this would lead to some phenomena being unproportionally weighted in the composite vulnerability index. Further, with this many variables, it is also more difficult to pinpoint and isolate potential socio-economic entry-points for reducing drought vulnerability. Hence, it might be worthwhile to analyze how the variables correlate and identify the most relevant indicators using e.g. PCA.

3. Fig. 6: The comparison between the drought risk indicator developed here and the drought hazards observed in EM-DAT is a relevant and nice addition to the study. However, visually it does not seem that the amount of observed drought hazards correlate with the risk indicator presented. I would recommend showing a scatter plot about this relationship (especially, since the authors refer to this section as a validation of the proposed drought risk indicator), at least for those areas where data exist for both sources, so that the reader can assess their agreement more easily.

4. The authors have assessed the risk of drought by combining the associated hazard, exposure and vulnerability components. However, the difference between hazard and exposure is currently not clearly stated and defined in the manuscript. For example, what are the drought exposure and hazard components used for deriving the results in

Figs 2 and 3. Further, it would be good to explicitly explain the exposure component in the text (exposure of what?), since also some of the vulnerability indicators could be viewed as being related to drought exposure (e.g. % of GDP from agriculture etc., rural population).

5. The GCWM was forced with monthly data, which were transformed into pseudo daily climate. As products that readily have daily records exist (e.g. AgMerra, ISI-MIP forcing), why they chose to use monthly forcing data?

6. Minor comment for structure: would be good to be consistent between methods and results in which order you present the results (method: rainfed, irrigated; results: irrigated, rainfed)

7. It would be worthwhile to cross-refer to Fig. 1 in describing the methods, as it would make the methods easier to understand. This would also bind Fig. 1 better to the rest of text, as now it is a bit isolated from it.

8. I would recommend tabulating also the other data than vulnerability indicators used for the study, so that the reader can get an understanding of the data more quickly and easily.

9. Page 5 Rows 116-117: The definition of the MIRCA-areas is a bit unclear.

10. Figures 2 and 3: The range for color scales of the figures should be the same, at least for the hazard and vulnerability figures. Currently, it is very difficult to assess the contribution of each component on the total risk factor, and it seems that the hazard component has a way stronger influence on the drought risk compared to vulnerability (the mapped patterns are essentially the same for hazard/exposure and risk).

11. Why hazard/exposure for rainfed is computed at national/sub-national level? Further, why these are aggregated to national level in analyses of for agricultural systems? These aggregations make it hard to compare the different results. It is of course ok to finally aggregate the results to country scale, but would be good show also the nonaggregated results for all the results.

12. Fig. 5: Would be good to have the y and x axes in same scale, not to give misleading impression of the results. And/Or you could show 1:1 line, and with that it would be easier to see in which countries risk irrigated agriculture is higher/lower than in rainfed agriculture

13. Page 7, Lines 170-173: Why is IH transformed logarithmically?
* * *

---

## Referee Comment (RC2) · Veit Blauhut (Referee) · 25 Sep 2019

[revised manuscript text omitted]
 for 412 MIRCA2000-units in subnational units for Argentina, Australia, Brazil, China, India, Indonesia and USA; and at a national scale elsewhere) or at country level. MIRCA2000 was also used to inform the models used in the hazard calculations about growing areas and growing periods of irrigated and rain-fed crops. The data set refers to the period centered around the year 2000; time series

120    information is not available at the global scale. To maximize the representativeness of the land use, the reference period and evaluation period used in this study were centered around the year 2000.

**2.1.1 Rain-fed agricultural systems**

The composite drought hazard indicator for rain-fed agriculture (*CH_RfAg*) was quantified based on the ratio of actual crop evapotranspiration (AET - in $m^3$ $day^{-1}$) to potential crop evapotranspiration (PET in $m^3$ $day^{-1}$), calculated for the evaluation

125    period 1980-2016 and compared to the reference period 1986-2015. PET quantifies the water requirement of the crop without water limitation while AET refers to the evapotranspiration under actual soil moisture conditions.

The GCWM was applied for 24 specific rain-fed crops and the two groups "others annual" and "others perennial" to calculate crop specific AET and PET on a daily time step. Together, the 24 crops and two crop groups cover all crop species distinguished by FAO in their database FAOSTAT. The sum of daily crop specific AET and PET was calculated for all crops and for each

130    year in the period 1980-2016 and aggregated to MIRCA2000-units. This procedure accounted for the differences in growing areas of the specific rain-fed crops across grid cells belonging to the same MIRCA2000 unit and therefore reflects the different exposure of specific crops in different parts of the MIRCA2000 unit to drought.

[revised manuscript text omitted]

Seven out of the ten countries with the highest overall drought risk are located on the African continent. However, Armenia,

Yemen and Hungary also possess high risk levels (Table 02). Botswana ranks as the country with the highest drought risk

mainly due to its high exposure combined with its relatively high vulnerability (S1).

365

In general, the countries that present higher drought risk have a high amount of exposed crops. Vulnerability varies among

them, with Yemen being the country with the highest vulnerability. The lack of coping capacity and social-ecological

susceptibility were determinant factors for countries like Yemen and Zimbabwe (S1). There were cases where countries such

as Namibia presented high socio-ecological susceptibility in contrast with high coping capacity, reducing its overall

370    vulnerability. The drought risk in countries such as Afghanistan and Venezuela that have in contrast limited coping capacities,

is notably higher (S1). The analysis also reveals that, although risk is currently zero in several countries (e.g. DRC, C.A.R., Uganda, etc.), this could rapidly change once these countries are affected by droughts given their very high vulnerability.

[revised manuscript text omitted]

However, despite these advancements, the presented analysis does have limitations. First, due to the lack of up to date land use data on irrigated vs. rain-fed agriculture at the global scale, the exposure analysis is based on MIRCA data from the year 2000 (Portmann et al., 2000). Given that cropping systems are subject to change, this adds uncertainty to the results. Drought exposure in large countries with variable climate conditions such as Russia and Canada needs to be viewed critically, since drought exposure is significantly higher in some parts of these countries when conducting the analysis at provincial or pixel level. For instance, China shows a high variation in exposure levels in the eastern and western parts of the country when

[revised manuscript text omitted]

---

## Author Comment (AC2) · 6 Nov 2019

Veit Blauhut Referee comment:

Dear authors,firstly, please excuse my delay submitting this report. secondly, congratulations to avery well written piece of work. The papers reads very fluent and only leaves space for few questions.

The authors are exploring agricultural drought risk on a global scale, comparing rainfed against irrigated agriculture in the frame of a conceptual model. I highly appreciate this unique attempt, especially the preceding expert survey. My major concern is a lack of validation e.g. with other global agricultural drought risk models and the "lacking"verification of the relevance of selected indicators. More data exists. Also quantitative approaches for validation exist. I please you to apply something that is not "visual comparison" (This really lowers the quality of your, apart from that, high quality paper)

Response: Many thanks for the positive overall summary. To our knowledge, the analysis presents the first attempt to assess drought risk for agricultural systems at the global scale. Existing global analysis focus on drought risk in more general terms (e.g. Carrao et al., 2016, Dilley et al., 2005). Hence, a direct statistical comparison might be misleading - due to the different foci of the studies. We have visually compared our results to the work done by Carrao et al. (2016) and mention this in the discussion. An in-depth comparison and validation, while needed, would be a paper of its own. We will add a few lines in the discussion mentioning this as an outlook for future research.

1.        Please    find    my    more    explicit    comments    in    the    PDF https://docs.google.com/document/d/19w7_cn6r4t3rKJxqq51H6I6veY6G5vZBLkgN-zUNbcs/edit

Response: Many thanks, kindly find our responses below.

[a] P1 L20-21: A glimps if the assessment is of general risk information or applicable for early warning would be nice.

Response: Thank you for the comment. We will make it more clear that the aim of this paper is to conduct a risk assessment for agricultural systems and not to focus on early warning.

[b] P2 L49: I assume that you know what are you talking about BUT: please consider to cleary define risk relevant terminologies. Many terms are interpreted differently by schools, scientific field or even authors. Maybe a list/table in the appendix could do the job.

Response: Thanks for the comment. This comment has been raised by the other reviewer as well. We follow the latest IPCC AR5 WGII (IPCC 2014) definitions of risk. We will add a line in the methods (chapter 2) on the risk concept that is used.

[c] P2 L65: Since you are setting a point on exposure I recommend to define this. Exposure is treated differently in drought risk community (from landuse to drought frequency)- thus a clarification is of need. Please see De Stefano et al. 2015 & Gonzales Tanago et al. 2016.

Response: Many thanks, this comment is also related to the previous comment on definitions and has also been raised by the other reviewer. We follow the IPCC (2014) definition. We will add a line in the methods (chapter 2.1) on how exposure is defined.

[d] P2 L65:Partly true. Dilley et al. 2005 and Li et al. 2017 also investigate at global scale. But using different/ and less vulnerability factors

Response: Thanks for the suggestion, we will add the paper from Dilley et al. (2005) to the introduction. The paper by Li et al. (2017) has a geographic focus on Northwest China. Hence it was not mentioned in the introduction. We decided not to add it since the analysis is not global.

[e] P3 L72-75: You might also bring their "validation" schemes into play?

Response: Thanks for the suggestion, Carrao et al. (2016) evaluated the robustness of their analysis to changes in indicator weights. Their model is based on an internal validation procedure that chooses the best model as the one giving regional vulnerability ranks that approximates the median of the ensemble among all models tested. Nevertheless, in this case the absence of reference data for performing an independent validation reduces the lack of effective testing options. A statistical validation of the sensitivity of the results towards changes in the input parameters (indicators, weights, normalization, aggregation methods), while needed, goes beyond the scope of this paper. We will add a few lines in the discussion to mention this as a possibility for future research.

[f ] P3 L77: I see the point, but not the matter for your research.

Response: A social-ecological systems (SES) perspective is relevant when assessing drought risk for agricultural systems, which by definition have a strong social and ecological coupling. This point was also highlighted in a recent assessment of social-ecological systems vulnerability of deltaic regions confronted by multiple hazards - including droughts - by Hagenlocher et al. (2018).

[g] P3 L82: I question why a conceptual model should be "better" to analyse agricultural drought risk then a global analysis based on crop- models (such as Li et al 2009, Yin et al. 2014, Zhang...). Please explore the caveats of outcome related vs. conceptual models with the background of your, now published, drought risk review.

Response: Here we used an integrated drought risk assessment approach based on the risk concept put forward by the IPCC WGII in their 5th Assessment Report (IPCC 2014). The advantage of such a conceptual model over an outcome oriented model (that estimates vulnerability indirectly by looking at losses) is that our approach allows for revealing drivers of risk (incl. vulnerability drivers) and hence entry points for vulnerability reduction whereas an outcome-based assessment approach does not allow for that. One strong advantage is that our approach provides comprehensive, aggregated, comparable and data-driven information on the actual vulnerability conditions and patterns at the global scale.

[h] P3 L82: What is an integrated drought risk assessment?

Response: An integrated drought risk assessment combines drought hazard, exposure, and vulnerability by bringing together data from different sources and disciplines. We will make that more clear in the text.

[i] P4 L100: ??

Response: The answer to this point is linked to the comment on P3 L77, which was already addressed.

[j] P6 L140-143: This indeed is a little arbitrary. Of cause, most thresholds are subjective like this one. And I actually do not have a better idea, bit maybe you could

provide some explanatory box/violine plots. Are there regional/ continental specifics? (appendix is fine)

Response: In total there are 37,265 grid cells of the size 0.5 x 0.5 degree containing rainfed cropland. With the threshold of 10% selected for the present study, no drought at all would be observed in the period 1980-2016 in 3,999 grid cells (10.7%), mainly and very humid or cool climate. The number of grid cells without any drought would increase to 11,879 (31.9%) when using a threshold of 20% and to 20,803 grid cells (55.8%) when using a threshold of 30%. We decided, therefore, to keep the threshold of 10% but will add some descriptive statistics on the effect of this threshold to the supplementary information.

[k] P7 L170: Why?

Response: Many thanks. This comment was already raised by "Reviewer 1" (see comment #13). IH describes the volume of irrigation water needed additionally in drought periods. In most grid cells these volumes are relatively small but there are also some grids with extremely high values. In 569 out of the 26,478 irrigated grid cells the additional irrigation water requirement per drought event is lower than 100 $m^3$; in 1,450 grids it is lower than 1,000 $m^3$. These are grids with very small irrigated areas. However, there are also 95 grids where the additional irrigation water requirement per drought event is larger than 100,000,000 $m^3$. The logarithmic transformation accounted for the value distribution.

[l] P8 L212-214: Again, please refer to Gonzales Tanago et al. 2016 and add some cons of this practise

Response: Thanks for the suggestion. We have carefully read the paper by Gonzales Tanago et al. (2016), and it was already cited in the initial version of our manuscript. However, the authors do not mention specific limitations of index-based approaches. Hence for this point we decided not to refer to the paper. But we will add some cons about this approach following suggestions by Naumann et al. (2018), Beccari (2016),

and de Sherbinin et al. (2019).

[m] P9 L231: Similarity? Did you test this in the frame f a similarity test? Or Did you do cross-correlations? Please be more specific and consider to show your pre-selection criteria/ similarity tests, cross correlations. (again, appendix is fine)

Response: Thank you for the comment. We agree that the sentence is not clear. The decision of which indicators to combine was not based on statistical similarity tests, but on "logical reasoning" due to what these indicators represent. For instance Agriculture (% of GDP) and Dependency on agriculture for livelihood (%) were combined under one income indicator and the variables GDP per capita, PPP and Population below the national poverty line (%) both refer to poverty and therefore combined in one integrated indicator. We will reframe the sentence and add which (and how) indicators were combined in the text.

[n] P11 L288: I'm aware that only few global datasets on drought impacts are available. With respect to the EM-DAT database regions like Europe are not well represented in the database. E.g. Spain and Northern Europe did not suffer a single drought event, Portugal maybe 2 ,etc. Thus, I question the reliability of this source and wonder why you did not compare to other global, maybe impact based, drought risk analysis.

Response: We agree that the number in EM-DAT depends highly on data availability and the size of the country is important. We discussed these limitations in the manuscript (lines 397-402; 492-502) and countries are mentioned as an example. Regarding the comparison with other drought risk analyses: To our knowledge, the analysis presents the first ever attempt to assess drought risk for agricultural systems at the global scale. Existing global analysis focus on drought risk in more general terms (e.g. Carrao et al., 2016, Dilley et al., 2005), and thus a direct comparison could be misleading due to the different foci. An in-depth comparison of existing global drought risk assessments goes beyond the scope of this paper. It could however be an interesting piece of work for future research. We will add this as an outlook to the discussion. We

also agree that calling this approach "validation" might be misleading, and will hence change it to "comparison".

[o] P11 L292: Coming back here (just read the end): Your validation is of "visual nature". With respect to your own research (review paper) please consider to at least try to get some numbers behind your statements. Naumann et al. 2014 showed an easy approach to compare/validate relative pattern of vulnerability. This could also be done with the results of Carao et al.This way you could have to "different" datasets for validation

Response: A direct numeric comparison is difficult due to the lack of an actual validation data set that represents the ground truth with adequately high spatial or temporal resolution. We decided to use EM-DAT because it systematically collects reports of drought events and drought impacts from various sources, including UN agencies, NGOs, insurance companies, research institutes and press agencies (lines 288-289). We will add more about the point the referee is raising on the discussion, and mention the advantages and the need to have an in-depth statistical validation of drought events for future research.

[p] P16 L379-381: This seems counter intuitive for me. E.g. a country for Iran is heavily depending on irrigation. The absurd overuse of groundwater within the last decade led to an extreme drop in groundwater levels, which in a next step, increases the vulnerability of the farmers. Of cause, this comes back to the lack of knowledge on the dynamics of vulnerability.

Response: We agree with the comment. We will add a line on vulnerability dynamics on the paragraph.

[q] P20 L455: Did not see to many ecological vulnerability factors, only "Terrestrial and marine protected areas", soil erosion and fertilizer could also be shifted to technical and landuse, or?I would recommend not to highlight your research as social-ecological. (but it might be a matter of taste)

Response: Thank you for the comment. Even if we moved them to another category, they still represent the environmental susceptibility of the country. Especially when assessing drought risk in the context of agricultural systems, which are by definition SES (Kloos and Renaud 2016), a SES perspective is of relevance. This point has been raised by the same reviewer before (see our response to comment P3 L77).

[r] P20 L461: I expected a more intense comparison

Response: Many thanks. Our analysis has a distinct focus on agricultural systems, while Carrão et al. (2016) present a more generic drought risk assessment at the global scale. An in-depth statistical comparison of the findings hence might even be misleading due to the different foci of the analyses. We have, however, conducted a visual comparison of our findings and their findings.

[s] P21 L493: But there are more data (modelled but..) Since you are not satisfied with EM-DAT, I do not understand why you did not use others.

Response: This point is related to the one in the [o] P11 L292. We decided to use EM-DAT because it systematically collects reports of drought events and drought impacts from various sources, including UN agencies, NGOs, insurance companies, research institutes and press agencies (lines 288-289). We will add more about the points the referee has been raising on the discussion, and mention the advantages and the need to have an in-depth statistical validation of drought events for future research.

References:

Beccari, B. (2016). A Comparative Analysis of Disaster Risk, Vulnerability and Resilience Composite Indicators. PLoS Currents. https://doi.org/10.1371/currents.dis.453df025e34b682e9737f95070f9b970

Carrão, H., Naumann, G., & Barbosa, P. (2016). Mapping global patterns of drought risk: An empirical framework based on sub-national estimates of hazard, exposure and vulnerability. Global Environmental Change, 39, 108-124.

https://doi.org/10.1016/j.gloenvcha.2016.04.012.

de Sherbinin, A., Bukvic, A., Rohat, G., Gall, M., McCusker, B., Preston, B., Apotsos, A., Fish, C., Kienberger, S., Muhonda, P., Wilhelmi, O., Macharia, D., Shubert, W., Sluizas, R., Tomaszewski, B. & Zhang, S. (2019). Climate vulnerability mapping: A systematic review and future prospects. Wiley Interdisciplinary Reviews: Climate Change. https://doi.org/10.1002/wcc.600

Dilley, M., Chen, R.S., Deichmann, U., Lerner-Lam, A.L. Arnold, M., Agew, J., Buys, P., Kjevstad, O., Lyon, B. & Yetman, G. (2005). Natural Disaster Hotspots: a Global Risk Analysis. World Bank Publications, Washington, DC: World Bank.

Hagenlocher, M., Renaud, F. G., Haas, S., & Sebesvari, Z. (2018). Vulnerability and risk of deltaic social-ecological systems exposed to multiple hazards. Science of The Total Environment, 631–632, 71–80. https://doi.org/10.1016/j.scitotenv.2018.03.013

IPCC: Climate Change (2014). Synthesis Report. Contribution of Working Groups I, II and III to the Fifth Assessment Report of the Intergovernmental Panel on Climate Change [Core Writing Team, R.K. Pachauri and L.A. Meyer (eds.)]. IPCC, Geneva, Switzerland, 151 pp.

Naumann, G., Carrão, H., & Barbosa, P. (2018). Indicators of Social Vulnerability to Drought. In A. Iglesias, D. Assimacopoulos, & H. A. J. Van Lanen (Eds.), Drought (pp. 111–125). https://doi.org/10.1002/9781119017073.ch6

---

## Author Response (AR1)

**NOTE:** This document includes (1) a **summary of the most relevant revisions** made in the manuscript, (2) a **point-by-point response** to the reviewer's comments, (2), as well as (3) the **revised manuscript**. Changes in the manuscript were made using "track changes", so that these can be identified easily. Page and line numbers in the point-by-point response refer to the final revised manuscript (clean version without track changes).

**Most relevant changes made in the manuscript**

- Key terms that build the basis for our analysis (i.e. exposure, vulnerability, hazard) are now clearly defined in the revised manuscript (building on latest IPCC definitions). Vulnerability and hazard were already defined in the initial version; we have now added a clear definition of exposure.

- The hazard/exposure analysis for rainfed systems was computed at grid-cell level (now the results for both systems, rainfed and irrigated, are available at grid-cell level)

- A table was added that provides an overview of the hazard and exposure indicators used in the analysis and their processed input data (Table 1)

- The drought risk assessment for both rainfed and irrigated agricultural systems was computed again with the new grid-cell data (for rainfed systems). Tables and figures related to it where changed accordingly

- The discussion was revised, further emphasizing opportunities, limitations, and potential future research needs

- A more in-depth comparison of our results and the results of the global risk assessment by Carrao et al. (2016) was added to the discussion chapter

**Point-by-point response to reviewer's comments**

**Referee #1 (Anonymous)**

The authors present a relevant and interesting manuscript, where they have studied and mapped composite drought risk at the global scale. For assessing agricultural drought risk, they have separated drought hazard/exposure in irrigated and rainfed cropping systems, and combined these hazard indicators with socio-ecological vulnerability. Finally, they have compared the obtained drought risk metric, with reported drought hazard events from EM-Dat. In general, I like and agree with the approach described in the study. My notes about the study are written below. Many thanks for the overall positive feedback.

**1**. I very much agree with looking into drought hazard for irrigated and rainfed cropping systems separately. However, the way these hazard indicators are combined, is potentially misleading. The hazard indicators are combined in a way that equalizes the weight of drought hazards in irrigated and rainfed cropping systems. However, as irrigated systems are, in general, more resilient to drought (irrigated systems can mitigate drought impacts by irrigation while rainfed systems cannot), equalizing the hazard associated with rainfed and irrigated systems, does not seem sensible. Further, if I understand correctly, based on the analyses, drought is more frequent in irrigated cropping systems compared to rainfed systems, which is not something that would be initially expected (Page 7, Lines 193-194). To make the methods comparable across rainfed and irrigated cropping systems, the authors could potentially define droughts for rainfed systems as for irrigated systems, but without the option to compensate the demand deficit by irrigation.

We agree that combining the indicators for rainfed and irrigated drought risk may be misleading, and we have highlighted this aspect already in the discussion section (**lines 434-437**). However, we don't agree that irrigated systems are in general more resilient than rainfed systems. The way how rainfed and irrigated systems mitigate drought is different. In rainfed systems crops are often cultivated in the wet season and soil conservation methods or water concentration are used to accumulate soil moisture in the cropped soils. Irrigated systems allow crop cultivation in arid regions or in the dry period of the year. For example, half of the global irrigated

land is located in arid and semi-arid regions (Siebert et al., 2015) and the majority of irrigation water requirement in South Asia is in the dry Rabi season (Biemans et al., 2016). However, these systems rely heavily on the functioning of the water supply infrastructure, which is often extremely complex. There are many reasons why water supply to irrigated fields often fails in practise, in particular during drought events. We account for these differences by using different indicators to calculate drought hazard for rainfed and irrigated systems.

We agree that our assumption of a similar weight for irrigated and rainfed hazard is questionable, but to quantify these weights would require a lot of region specific information that is not available at global scale (we mention this aspect in the discussion).

We want to highlight that the reasoning for the calculation of the total risk in this manuscript was less to support comparisons across countries but to account for the different extend of irrigated and rainfed systems within the specific countries. There are countries in which crop production is completely rainfed and countries in which all crops are irrigated so that only the risk for the rainfed or irrigated systems are relevant. Except from these extremes, crop production in most countries is either predominantly irrigated or predominantly rainfed. We account for this by calculating total risk as the harvested area weighted mean of the rainfed versus irrigated drought risk. We added some lines to the revised version of the manuscript to explain this reasoning and to support the interpretation of the total risk maps **(P19, L437-443)**.

**2**. The vulnerability assessment includes a high number of indicators. Although, the authors have excluded variables that have >0.9 correlation, many of the indicators are still most likely highly correlated. Considering the method used for calculating the vulnerability metric, this would lead to some phenomena being unproportionally weighted in the composite vulnerability index. Further, with this many variables, it is also more difficult to pinpoint and isolate potential socio-economic entry-points for reducing drought vulnerability. Hence, it might be worthwhile to analyze how the variables correlate and identify the most relevant indicators using e.g. PCA.

Thanks for the comment. For the multicollinearity analysis we follow a standard approach for composite indicator construction, as e.g. described by OECD in their Handbook on Composite Indicator Construction (OECD, 2008). Indeed the multicollinearity analysis has revealed that several of the indicators are highly correlated with correlations >0.7. As mentioned in the methods section (lines **255-256**) we have decided to keep highly correlated indicators when they present different drivers of vulnerability (and hence different entry points for vulnerability reduction). We added some lines in the discussion **(P20, L464-466)**, where we mention this as a potential limitation of our work.

Regarding PCA, we understand that there are different approaches to identify a final set of weighted vulnerability indicators, incl. statistical approaches (e.g. PCA) and expert-based approaches. Here, an expert-based approach is applied where indicators were identified based on a review of the literature and their relevance was evaluated based on expert judgment (n = 78 experts participated in the weighting). We added a few lines in the discussion that statistical approaches (e.g. based on PCA) could have been an alternative to the expert-based approach used here, and indicate this as an outlook for future research to compare the findings of statistical and expert-based approaches (**P20, L471-476**). Such a comparison (PCA vs. expert-based approach) was conducted by Hagenlocher et al. (2013), who did not find significant impacts of the choice of the approach on the final vulnerability index.

**3**. Fig. 6: The comparison between the drought risk indicator developed here and the drought hazards observed in EM-DAT is a relevant and nice addition to the study. However, visually it does not seem that the amount of observed drought hazards correlate with the risk indicator presented. I would recommend showing a scatter plot about this relationship (especially, since the authors refer to this section as a validation of the proposed drought risk indicator), at least for those areas where data exist for both sources, so that the reader can assess their agreement more easily.

Thank you for the comment, a scatter plot will not improve the visualization of the results just because EM-DAT is not based on physical parameters to record droughts. The number in EM-DAT depends highly on data availability (some countries in Africa are frequently affected by drought, but this is not recorded consistently) and the size of a country is important (larger countries often indicate a higher number of drought events, e.g. China, USA, Russia). This is described in the manuscript (lines **402-408; 523-528**) and countries are mentioned as an example.

**4**. The authors have assessed the risk of drought by combining the associated hazard, exposure and vulnerability components. However, the difference between hazard and exposure is currently not clearly stated and defined in the manuscript. For example, what are the drought exposure and hazard components used for deriving the results in Figs 2 and 3. Further, it would be good to explicitly explain the exposure component in the text (exposure of what?), since also some of the vulnerability indicators could be viewed as being related to drought exposure (e.g. % of GDP from agriculture etc., rural population).

Thanks for the comment, following the IPCC (2014) definition of exposure as "elements located in areas that can be potentially affected by hazards", exposure in our analysis is directly related to the hazard. As described in section 2.1, we used rainfed and irrigated croplands according to the Monthly Irrigated and Rainfed Cropping Areas (MIRCA2000) dataset (Portmann et al., 2010) as the exposed element. We add the definition of exposure in the methods section (2.1 chapter, **P5, L115-116**). GDP from agriculture was included as a vulnerability indicator since countries with high dependency on agriculture, are more vulnerable to droughts; this indicator was also the most relevant indicator ranked by the experts.

**5**. The GCWM was forced with monthly data, which were transformed into pseudo daily climate. As products that readily have daily records exist (e.g. AgMerra, ISI-MIPforcing), why they chose to use monthly forcing data?

We agree that the way how pseudo-daily climate is generated from monthly input data represents a source of uncertainty. However, since drought is an event that develops slowly, we are confident that our basic findings are not affected by this limitation. For the methods how drought hazard is calculated in our study it does not matter much whether a rainfall event is a few days earlier or later.

We are aware of alternative data sets that could be used as climate input and in fact we started to process global reanalysis data to explore the potential of using these daily data in GCWM and WaterGAP as climate input. However, the products mentioned by the reviewer have own limitations. AgMerra is only available for the period until 2010, similar to some other climate data sets used by ISIMIP. Other data sets do not provide all the variables needed to calculate potential evapotranspiration using the Penman-Monteith method. WFD and some other products, for example, do not provide daily maximum and daily minimum temperatures which are essential to quantify accurately the vapor pressure deficit. Finally, ISIMIP input data have a resolution of 0.5 by 0.5 degree while GCWM is running on a native 5 arc minute resolution (0.0833 degree). Because of these challenges, we decided to use the well established CRU monthly climate input for the present paper and refer to future activities and future studies in which we will explore the use of daily climate input data.

**6**. Minor comment for structure: would be good to be consistent between methods and results in which order you present the results (method: rainfed, irrigated; results: irrigated, rainfed)

Thank you, we changed it accordingly.

**7**. It would be worthwhile to cross-refer to Fig. 1 in describing the methods, as it would make the methods easier to understand. This would also bind Fig. 1 better to the rest of text, as now it is a bit isolated from it.

Thank you for the comment, we changed it accordingly and add the fig. 01 through the text on lines **131, 139,177 and 280.**

**8**. I would recommend tabulating also the other data than vulnerability indicators used for the study, so that the reader can get an understanding of the data more quickly and easily.

Thanks for the suggestion. For a better understanding of which data was used for the hazard/exposure analysis we added an explanatory table **(Table 01)** with datasets and sources in section 2.1 **(P5, L125-127).**

**9**. Page 5 Rows 116-117: The definition of the MIRCA-areas is a bit unclear.

Thank you, we re-run the rainfed hazard/exposure assessment at pixel level, therefore it wasbe no need to describe the MIRCA-areas anymore, the text was revised to omit the lines that describe it (e.g. lines in the previous manuscript **116-118, 131-133, 136, 186, 190, 202-204**) and emphasis was placed on the analysis at pixel level.

**10**. Figures 2 and 3: The range for color scales of the figures should be the same, at least for the hazard and vulnerability figures. Currently, it is very difficult to assess the contribution of each component on the total risk factor, and it seems that the hazard component has a way stronger influence on the drought risk compared to vulnerability (the mapped patterns are essentially the same for hazard/exposure and risk).

We agree and are aware of this as a limitation from the values perspective, since the number depends on the set of variables and the methodology used. The different components have the same color scheme to better differentiate the minimum and maximum values of each one. However, the advantage of showing values is that

readers can reproduce why the risk map is colored like it is. We add how the classes were made in the caption of the figures to help the readers to make their own categorical classification while keeping the numerical info.

**11**. Why hazard/exposure for rainfed is computed at national/sub-national level? Further, why these are aggregated to national level in analyses of for agricultural systems? These aggregations make it hard to compare the different results. It is of course ok to finally aggregate the results to country scale, but would be good show also the non-aggregated results for all the results.

Thank you for the comment. The rainfed hazard/exposure was computed at grid cell level and the figures have been changed accordingly **(Fig. 02 and 03)**.

**12**. Fig. 5: Would be good to have the y and x axes in same scale, not to give misleading impression of the results. And/Or you could show 1:1 line, and with that it would be easier to see in which countries risk irrigated agriculture is higher/lower than in rainfed agriculture

Thanks for the comment. We changed the axes to normalized risk scores **(Fig. 05)**.

**13.** Page 7, Lines 170-173: Why is IH transformed logarithmically?

IH describes the volume of irrigation water needed additionally in drought periods. In most grid cells these volumes are relatively small but there are also some grids with extremely high values. In 569 out of the 26,478 irrigated grid cells the additional irrigation water requirement per drought event is lower than 100 m³; in 1,450 grids it is lower than 1,000 m³. These are grids with very small irrigated areas. However, there are also 95 grids where the additional irrigation water requirement per drought event is larger than 100,000,000 m³. The logarithmic transformation accounted for the specific value distribution.

**Referee #2 (Dr. Veit Blauhut)**

Dear authors, firstly, please excuse my delay submitting this report. secondly, congratulations to a very well written piece of work. The papers reads very fluent and only leaves space for few questions.

Many thanks for the positive overall summary.

The authors are exploring agricultural drought risk on a global scale, comparing rainfed against irrigated agriculture in the frame of a conceptual model. I highly appreciate this unique attempt, especially the preceding expert survey. My major concern is a lack of validation e.g. with other global agricultural drought risk models and the "lacking" verification of the relevance of selected indicators. More data exists. Also quantitative approaches for validation exist. I please you to apply something that is not "visual comparison" (This really lowers the quality of your, apart from that, high quality paper)

To our knowledge, the analysis presents the first attempt to assess drought risk for agricultural systems at the global scale. Existing global analysis focus on drought risk in more general terms (e.g. Carrao et al., 2016, Dilley et al., 2005). Hence, a direct statistical comparison might be misleading - due to the different foci of the studies. We have visually compared our results to the work done by Carrao et al. (2016) and mention this in the discussion. An in-depth comparison and validation, while needed, would be a paper of its own. We added a few lines in the discussion **(P22, L531-534)** mentioning this as an outlook for future research.

Please find my more explicit comments in the PDF
https://docs.google.com/document/d/19w7_cn6r4t3rKJxqq51H6I6veY6G5vZBLkgN-zUNbcs/edit
Many thanks, kindly find our responses below.

[a] P1 L20-21: A glimps if the assessment is of general risk information or applicable for early warning would be nice.

Thank you for the comment. The aim of this paper is to conduct a risk assessment for agricultural systems and not to focus on early warning/forecasting drought hazards. We believe that this is sufficiently clear from the abstract. No changes were made.

[b] P2 L49: I assume that you know what are you talking about BUT: please consider to cleary define risk relevant terminologies. Many terms are interpreted differently by schools, scientific field or even authors. Maybe a list/table in the appendix could do the job.

Thanks for the comment. This comment has been raised by the other reviewer as well. We follow the latest IPCC AR5 WGII (IPCC 2014) definitions of risk. We added a line **(P5, L115-116)** in the methods (chapter 2) on the risk concept that is used. Vulnerability and hazard were already defined **(P8, L209-2011; P4, L105-106)**

[c] P2 L65: Since you are setting a point on exposure I recommend to define this. Exposure is treated differently in drought risk community (from landuse to drought frequency)- thus a clarification is of need. Please see De Stefano et al. 2015 & Gonzales Tanago et al. 2016.

Many thanks, this comment is also related to the previous comment on definitions and has also been raised by the other reviewer. We follow the IPCC (2014) definition. We added a line in the methods (chapter 2.1; **P5, L115-116)** on how exposure is defined.

[d] P2 L65:Partly true. Dilley et al. 2005 and Li et al. 2017 also investigate at global scale. But using different/ and less vulnerability factors

Thanks for the suggestion, we added the paper from Dilley et al. (2005) to the introduction **(P3, L65)**. The paper by Li et al. (2017) has a geographic focus on Northwest China. Hence it was not mentioned in the introduction. We decided not to add it since the analysis is not global.

[e] P3 L72-75: You might also bring their "validation" schemes into play?

Thanks for the suggestion, Carrao et al. (2016) evaluated the robustness of their analysis to changes in indicator weights. Their model is based on an internal validation procedure that chooses the best model as the one giving regional vulnerability ranks that approximates the median of the ensemble among all models tested. Nevertheless, in this case the absence of reference data for performing an independent validation reduces the lack of effective testing options. A statistical validation of the sensitivity of the results towards changes in the input parameters (indicators, weights, normalization, aggregation methods), while needed, goes beyond the scope of this paper.

[f] P3 L77: I see the point, but not the matter for your research.

A social-ecological systems (SES) perspective is relevant when assessing drought risk for agricultural systems, which by definition have a strong social and ecological coupling. This point was also highlighted in a recent assessment of social-ecological systems vulnerability of deltaic regions confronted by multiple hazards - including droughts - by Hagenlocher et al. (2018).

[g] P3 L82: I question why a conceptual model should be "better" to analyse agricultural drought risk then a global analysis based on crop- models (such as Li et al 2009, Yin et al. 2014, Zhang...). Please explore the caveats of outcome related vs. conceptual models with the background of your, now published, drought risk review.

Here we used an integrated drought risk assessment approach based on the risk concept put forward by the IPCC WGII in their 5th Assessment Report (IPCC 2014). The advantage of such a conceptual model over an outcome oriented model (that estimates vulnerability indirectly by looking at losses) is that our approach allows for revealing drivers of risk (incl. vulnerability drivers) and hence entry points for vulnerability reduction whereas an outcome-based assessment approach does not allow for that. One strong advantage is that our approach provides comprehensive, aggregated, comparable and data-driven information on the actual vulnerability conditions and patterns at the global scale.

[h] P3 L82: What is an integrated drought risk assessment?

An integrated drought risk assessment combines drought hazard, exposure, and vulnerability by bringing together data from different sources and disciplines. We make that more clear in the text **(P3 L82-83)**.

[i] P4 L100: ??

The answer to this point is linked to the comment on [f], which was already addressed.

[j] P6 L140-143: This indeed is a little arbitrary. Of cause, most thresholds are subjective like this one. And I actually do not have a better idea, bit maybe you could provide some explanatory box/violine plots. Are there regional/ continental specifics? (appendix is fine)

In total there are 37,265 grid cells of the size 0.5 x 0.5 degree containing rainfed cropland. With the threshold of 10% selected for the present study, no drought at all would be observed in the period 1980-2016 in 3,999 grid cells (10.7%), mainly and very humid or cool climate. The number of grid cells without any drought would

increase to 11,879 (31.9%) when using a threshold of 20% and to 20,803 grid cells (55.8%) when using a threshold of 30%. We decided, therefore, to keep the threshold of 10% we add some descriptive statistics on the effect of this threshold to the supplementary information **(Supplementary material (S5))**.

[k] P7 L170: Why?
Many thanks. This comment was already raised by "Reviewer 1" (see comment #13). IH describes the volume of irrigation water needed additionally in drought periods. In most grid cells these volumes are relatively small but there are also some grids with extremely high values. In 569 out of the 26,478 irrigated grid cells the additional irrigation water requirement per drought event is lower than 100 m³; in 1,450 grids it is lower than 1,000 m³. These are grids with very small irrigated areas. However, there are also 95 grids where the additional irrigation water requirement per drought event is larger than 100,000,000 m³. The logarithmic transformation accounted for the value distribution.

[l] P8 L212-214: Again, please refer to Gonzales Tanago et al. 2016 and add some cons of this practise
Thanks for the suggestion. We have carefully read the paper by Gonzales Tanago et al. (2016), and it was already cited in the initial version of our manuscript. However, the authors do not mention specific limitations of index-based approaches. Hence for this point we decided not to refer to the paper. But we add some cons about this approach following suggestions by Naumann et al. (2018) and Beccari (2016) **(P8, L215 and P20, L461-464).**

[m] P9 L231: Similarity? Did you test this in the frame f a similarity test? Or Did you do cross-correlations? Please be more specific and consider to show your pre- selection criteria/ similarity tests, cross correlations. (again, appendix is fine)
Thank you for the comment. We agree that the sentence is not clear. The decision of which indicators to combine was not based on statistical similarity tests, but on "logical reasoning" due to what these indicators represent. For instance Agriculture (% of GDP) and Dependency on agriculture for livelihood (%) were combined under one income indicator and the variables GDP per capita, PPP and Population below the national poverty line (%) both refer to poverty and therefore combined in one integrated indicator. We have reframed the sentence and added which (and how) indicators were combined in the **P9, lines 234-237**.

[n] P11 L288: I'm aware that only few global datasets on drought impacts are available. With respect to the EM-DAT database regions like Europe are not well represented in the database. E.g. Spain and Northern Europe did not suffer a single drought event, Portugal maybe 2 ,etc. Thus, I question the reliability of this source and wonder why you did not compare to other global, maybe impact based, drought risk analysis.
We agree that the number in EM-DAT depends highly on data availability and the size of the country is important. We discussed these limitations in the manuscript (lines **402-408; 523-528**) and countries are mentioned as an example. Regarding the comparison with other drought risk analyses: To our knowledge, the analysis presents the first ever attempt to assess drought risk for agricultural systems at the global scale. Existing global analysis focus on drought risk in more general terms (e.g. Carrao et al., 2016, Dilley et al., 2005), and thus a direct comparison could be misleading due to the different foci. An in-depth comparison of existing global drought risk assessments goes beyond the scope of this paper. It could however be an interesting piece of work for future research. We added this as an outlook to the discussion **(P22, L531-534).** We also agree that calling this approach "validation" might be misleading, and hence change it to "comparison".

[o] P11 L292: Coming back here (just read the end): Your validation is of "visual nature". With respect to your own research (review paper) please consider to at least try to get some numbers behind your statements. Naumann et al. 2014 showed an easy approach to compare/validate relative pattern of vulnerability. This could also be done with the results of Carao et al.This way you could have to "different" datasets for validation
A direct numeric comparison is difficult due to the lack of an actual validation data set that represents the ground truth with adequately high spatial or temporal resolution. We decided to use EM-DAT because it systematically collects reports of drought events and drought impacts from various sources, including UN agencies, NGOs, insurance companies, research institutes and press agencies (**lines 291-292**). We added more about the point the referee is raising on the discussion, and mention the advantages and the need to have an in-depth statistical validation of drought events for future research **(P22, L531-534)**.

[p] P16 L379-381: This seems counter intuitive for me. E.g. a country for Iran is heavily depending on irrigation. The absurd overuse of groundwater within the last decade led to an extreme drop in groundwater levels, which in a next step, increases the vulnerability of the farmers. Of cause, this comes back to the lack of knowledge on the dynamics of vulnerability.
We agree with the comment. We added a line on vulnerability dynamics on the paragraph **(P16, L380)**.

[q] P20 L455: Did not see to many ecological vulnerability factors, only "Terrestrial and marine protected areas", soil erosion and fertilizer could also be shifted to technical and landuse, or?I would recommend not to highlight your research as social-ecological. (but it might be a matter of taste)

Thank you for the comment. Even if we moved them to another category, they still represent the environmental susceptibility of the country. Especially when assessing drought risk in the context of agricultural systems, which are by definition SES (Kloos and Renaud 2016), a SES perspective is of relevance. This point has been raised by the same reviewer before (see our response to comment [f]).

[r] P20 L461: I expected a more intense comparison

Many thanks. Our analysis has a distinct focus on agricultural systems, while Carrão et al. (2016) present a more generic drought risk assessment at the global scale. An in-depth statistical comparison of the findings hence might even be misleading due to the different foci of the analyses. We have, however, conducted a visual comparison of our findings and their findings. We added a paragraph on the discussion about the comparison between Carrão et al. (2016) and our paper (**P21, L478-492**).

[s] P21 L493: But there are more data (modelled but..) Since you are not satisfied with EM-DAT, I do not understand why you did not use others.

This point is related to the one in the [o]. We decided to use EM-DAT because it systematically collects reports of drought events and drought impacts from various sources, including UN agencies, NGOs, insurance companies, research institutes and press agencies (lines **291-292**). We added more about the points the referee has been raising on the discussion, and mention the advantages and the need to have an in-depth statistical validation of drought events for future research (**P22, L531-534).**

**References (used in the point-by-point response to support our arguments):**

Beccari, B. (2016). A Comparative Analysis of Disaster Risk, Vulnerability and Resilience Composite Indicators. PLoS Currents. https://doi.org/10.1371/currents.dis.453df025e34b682e9737f95070f9b970

Biemans, H., Siderius, C., Mishra, A. & Ahmad, B., 2016. Crop-specific seasonal estimates of irrigation-water demand in South Asia. Hydrology and Earth System Sciences 20, 1971-1982.

Carrão, H., Naumann, G., & Barbosa, P. (2016). Mapping global patterns of drought risk: An empirical framework based on sub-national estimates of hazard, exposure and vulnerability. Global Environmental Change, 39, 108-124. https://doi.org/10.1016/j.gloenvcha.2016.04.012.

Dilley, M., Chen, R.S., Deichmann, U., Lerner-Lam, A.L. Arnold, M., Agew, J., Buys, P., Kjevstad, O., Lyon, B. & Yetman, G. (2005). Natural Disaster Hotspots: a Global Risk Analysis. World Bank Publications, Washington, DC: World Bank.

Hagenlocher, M., Delmelle, E., Casas, I., & Kienberger, S. (2013). Assessing socioeconomic vulnerability to dengue fever in Cali, Colombia: Statistical vs expert-based modeling. International Journal of Health Geographics, 12(1), 36. https://doi.org/10.1186/1476-072X-12-36

Hagenlocher, M., Renaud, F. G., Haas, S., & Sebesvari, Z. (2018). Vulnerability and risk of deltaic social-ecological systems exposed to multiple hazards. Science of The Total Environment, 631–632, 71–80. https://doi.org/10.1016/j.scitotenv.2018.03.013

IPCC: Climate Change (2014). Synthesis Report. Contribution of Working Groups I, II and III to the Fifth Assessment Report of the Intergovernmental Panel on Climate Change [Core Writing Team, R.K. Pachauri and L.A. Meyer (eds.)]. IPCC, Geneva, Switzerland, 151 pp.

Naumann, G., Carrão, H., & Barbosa, P. (2018). Indicators of Social Vulnerability to Drought. In A. Iglesias, D. Assimacopoulos, & H. A. J. Van Lanen (Eds.), Drought (pp. 111–125). https://doi.org/10.1002/9781119017073.ch6

OECD (Ed.). (2008). Handbook on constructing composite indicators: Methodology and user guide. Paris: OECD.

Portmann, F. T., Siebert, S., & Döll, P. (2010). MIRCA2000-Global monthly irrigated and rainfed crop areas around the year 2000: A new high-resolution data set for agricultural and hydrological modeling: MONTHLY IRRIGATED AND RAINFED CROP AREAS. Global Biogeochemical Cycles, 24(1), n/a-n/a. https://doi.org/10.1029/2008GB003435

Siebert, S., Kummu, M., Porkka, M., Döll, P., Ramankutty, N. & Scanlon, B.R., 2015. A global data set of the extent of irrigated land from 1900 to 2005. Hydrology and Earth System Sciences 19, 1521-1545

[revised manuscript text omitted]

295 country level) (rain-fed agricultural system). At pixel level, the presence of hazard and vulnerability point to a certain drought risk, independent of how much crop area is contained in the specific pixel. At aggregated level, the different crop areas in the specific pixels must be considered; therefore exposure was calculated as harvested area weighted mean of the pixel level hazard and then multiplied with vulnerability to calculate drought risk at country level.

300 The total drought risk score for irrigated and rain-fed systems combined ($DRI_{tot}$) is derived by multiplying the exposure of the whole cropping system $Exp_{tot}$ (Equation 02) with the vulnerability index $VI$.

**2.3 ComparisonValidation against drought impact data**

The outcomes of the risk assessment for irrigated and rain-fed systems combined ($DRI_{tot}$) were comparvalidated against impact data from the international Emergency Events Database (EM-DAT) of the Centre for Research on the Epidemiology of

305 Disasters (CRED) using visual correlation (Fig. 06). EM-DAT systematically collects reports of drought events and drought impacts from various sources, including UN agencies, NGOs, insurance companies, research institutes and press agencies. Here, the number of drought events within the period 1980-2016 was used as an input for the comparisonvalidation. Thereby, a drought event is registered in EM-DAT when at least one of the following criteria applies: 10 or more people dead; 100 or more people affected; declaration of a state of emergency or a call for international assistance.

**310 3 Results**

This section presents the results of the global drought risk assessment for agricultural systems (irrigated and rain-fed) at pixel level (Fig. 02 and 03) and for the total risk of both systems combined at national resolution (Fig. 04). The drought risk for irrigated and rainfed agricultural systems is presented at 0.5 degrees (Fig. 02) and at national resolution (Fig. 04)., while drought risk for rain-fed systems is presented for MIRCA2000 units (at sub-national resolution for USA, Brazil, Argentina,

315 China, India, Australia and Indonesia; national resolution elsewhere (Fig. 03); and at national resolution (Fig. 04). Drought risk for all crops (irrigated and rain-fed) is shown at national resolution (Fig. 04) and for MIRCA2000 units (Fig. 06). The patterns colored dark red show high levels of the different risk components, while the dark blue colors reflects low scores of the different risk components.

320

**3.1 Drought risk for irrigated agricultural systems**

[revised manuscript text omitted]

Seven out of the ten countries with the highest overall drought risk are located on the African continent. However, Kosovo, East Timor and Kazakhstan Armenia, Yemen and Hungary also possess high risk levels (Table 032). ZimbabweBotswana ranks as the country with the highest drought risk mainly due to its high exposure combined with its relatively high vulnerability (S1).

[revised manuscript text omitted]

**(a)**

RISK-TOTAL (Country-level)

No crop
No data

High
0.871
0.535
0.490
0.444
0.398
0.353
0.304
0.251
0.198
0.145
0.092
Low
0.008

**(b)**

Number of drought events in EM-DAT

No drought occurred or
not registered in EM-DAT

High
10-31
7-9
5-6
4
2
Low

[Figure]

**Fig 06.** Comparison of total risk against drought impact data

**4 Discussion**

The present study performs, for the first time, a separate global drought risk analysis for irrigated and rain-fed cropping
systems, including regions that indicate a high vulnerability to droughts and are particularly exposed. In previous assessments,
the share of irrigated cropland was either ignored or considered as a vulnerability indicator (Carrão et al., 2016). The drought
hazard analysis is based on three indicators: streamflow drought hazard (*SH*), abnormally high irrigation water requirement
(*IH*), and a composite drought hazard indicator for rain-fed agriculture (*CH_RfAg*), which quantify drought as a deviation from
normal conditions consistent with common definitions. In agreement with the results for drought hazard obtained by Carrão et
al. (2016), the largest drought hazard is obtained for arid and semi-arid regions such as northern and southern Africa, northern
Mexico, along the coastline of Peru and Chile, the Arabian Peninsula and Mongolia for rain-fed systems,
Italy, Turkey and Western Mexico for irrigated systems, and the western USA, northeast Brazil, western Argentina, central
Asia, Middle East countries, western India, northern China and southern Australia for both irrigated and rain-fed systems. ~~For
irrigated systems this includes Italy, Turkey and Western Mexico, whilst for both irrigated and rain-fed systems this represents

[revised manuscript text omitted]